# Natural variation in salt-induced changes in root:shoot ratio reveals SR3G as a negative regulator of root suberization and salt resilience in *Arabidopsis*

Maryam Rahmati Ishka[1], Hayley Sussman[1], Yunfei Hu[2], Mashael Daghash Alqahtani[3], Eric Craft[4], Ronell Sicat[5], Minmin Wang[6], Li'ang Yu[1], Rachid Ait-Haddou[7], Bo Li[2], Georgia Drakakaki[6], Andrew DL Nelson[1], Miguel Pineros[4], Arthur Korte[8], Łukasz Jaremko[9], Christa Testerink[10], Mark Tester[3], Magdalena M Julkowska[1,3]*

[1]Boyce Thompson Institute, Ithaca, United States; [2]School of Life Sciences, Lanzhou University, Lanzhou, China; [3]Center for Desert Agriculture, King Abdullah University of Science and Technology, Thuwal, Saudi Arabia; [4]USDA-ARS, Ithaca, United States; [5]Visualization Core Lab, King Abdullah University of Science and Technology, Thuwal, Saudi Arabia; [6]University of California, Davis, Davis, United States; [7]Department of Mathematics, King Fahd University of Petroleum and Minerals, Dhahran, Saudi Arabia; [8]Julius-von-Sachs-Institute and Center for Computational and Theoretical Biology, Julius Maximilian University, Wuerzburg, Germany; [9]King Abdullah University of Science and Technology, Thuwal, Saudi Arabia; [10]Wageningen University & Research, Wageningen, Netherlands

*For correspondence: mmj55@cornell.edu

## eLife Assessment

Through cellular, developmental, and physiological analysis, this **valuable** study identifies a gene that regulates the relative growth of roots and shoots under salt stress. The holistic approach taken provides **convincing** evidence that this member of a larger tandemly duplicated gene family together with an upstream regulator contributes to salt tolerance. The manuscript will be of interest to plant biologists studying mechanisms of abiotic stress tolerance.

**Abstract** Soil salinity is one of the major threats to agricultural productivity worldwide. Salt stress exposure alters root and shoots growth rates, thereby affecting overall plant performance. While past studies have extensively documented the effect of salt stress on root elongation and shoot development separately, here we take an innovative approach by examining the coordination of root and shoot growth under salt stress conditions. Utilizing a newly developed tool for quantifying the root:shoot ratio in agar-grown *Arabidopsis* seedlings, we found that salt stress results in a loss of coordination between root and shoot growth rates. We identify a specific gene cluster encoding domain-of-unknown-function 247 (DUF247), and characterize one of these genes as Salt Root:shoot Ratio Regulator Gene (SR3G). Further analysis elucidates the role of SR3G as a negative regulator of salt stress tolerance, revealing its function in regulating shoot growth, root suberization, and sodium accumulation. We further characterize that *SR3G* expression is modulated by *WRKY75* transcription factor, known as a positive regulator of salt stress tolerance. Finally, we show that the salt stress sensitivity of *wrky75* mutant is completely diminished when it is combined with *sr3g* mutation. Together, our results demonstrate that utilizing root:shoot ratio as an architectural feature leads to

the discovery of a new stress resilience gene. The study's innovative approach and findings not only contribute to our understanding of plant stress tolerance mechanisms but also open new avenues for genetic and agronomic strategies to enhance crop environmental resilience.

## Introduction

Salt stress is predominant in the dryland areas where the evaporation rate exceeds water input. As all water contains dissolved ions, the prolonged exposure to drought stress results in increased accumulation of salts in the upper soil layers (*Abrol et al., 1988*; *Tramberend, 2021*; *Richards, 1969*). Dryland areas of South America, southern and western Australia, Mexico, the southwest United States, and South Africa were previously identified as primary hotspots for soil salinization (*Hassani et al., 2021*). As fresh water becomes scarcer, understanding the mechanisms of salt stress resilience becomes of paramount importance in the context of global agricultural productivity and sustainability.

Salt stress detrimentally affects plant growth and development, decreasing the activity of the meristem (*West et al., 2004*). Studies utilizing main root elongation as an indicator of salt tolerance have led to identification of many crucial components of salt stress signaling and tolerance, including genes involved in salt-overly sensitive signaling (SOS) pathway, jasmonate, ethylene, abscisic acid (ABA), and reactive oxygen species (ROS) in glycophytes and halophytes alike (*Fu et al., 2019*; *Geng et al., 2013*; *Hu et al., 2023*; *Şekerci et al., 2023*; *Valenzuela et al., 2016*; *Wu et al., 1996*). Additionally, the negative halotropism response, that describes a directional growth of the main root away from the salt gradient, was used to identify a wide range of molecular components regulating the auxin redistribution and cell-wall remodeling, to actively avoid high salt (*Deolu-Ajayi et al., 2019*; *Galvan-Ampudia et al., 2013*; *Korver et al., 2020*; *Yu et al., 2022*; *Zou et al., 2022*). The development of lateral roots is affected by salinity, and the differential effect on main and lateral roots within specific genotypes can result in substantial reprogramming of root system architecture (*Julkowska et al., 2017*; *Julkowska et al., 2014*). The shoot development is impacted by salt stress, as the salt impacts the plant transpiration and photosynthetic efficiency instantaneously (*Awlia et al., 2021*; *Awlia et al., 2016*; *Julkowska et al., 2016*). Many forward genetic studies were highly successful in associating natural variation in root and shoot growth with allelic variation in gene coding and promoter regions, thereby identifying potential new target traits for improved stress resilience (*Awlia et al., 2021*; *Julkowska et al., 2016*; *Julkowska et al., 2017*). However, these studies typically focus on either above or below ground tissue.

The coordination of root and shoot growth rate is an important aspect of seedling establishment, and becomes even more critical under abiotic stress (*Bacher et al., 2022*; *Chen et al., 2022*; *Karcher et al., 2008*; *Wang et al., 2018*; *Xu et al., 2015*). The importance of increased root:shoot ratio, maximizing water scavenging while limiting transpiration, is well studied under drought stress (*Asch et al., 2005*; *Chen et al., 2022*; *Karcher et al., 2008*; *Uga et al., 2013*). However, the effect of salt stress on root:shoot ratio is yet to be described, and its contribution to salt stress remains unknown. Here, we developed a tool to quantify root:shoot ratio from agar-grown *Arabidopsis* seedlings, and re-analyzed the natural diversity panel of *Arabidopsis* exposed to salt stress (*Julkowska et al., 2017*). We have observed that while salt stress does not elicit a clear response in root:shoot ratio, it results in a loss of coordination of root and shoot growth rates. The locus associated with salt-induced changes in root:shoot ratio consisted of a gene cluster encoding domain-of-unknown-function 247 (DUF247), with majority of SNPs clustering into one specific gene (At3g50160), that we named as Salt Root:shoot Ratio Regulator Gene (SR3G). Further characterization of SR3G revealed its function as a negative regulator of salt stress resilience, through regulation of root suberization, shoot growth, and $Na^+$ accumulation. We found WRKY75 transcription factor, a positive regulator of salt stress tolerance, acts upstream of SR3G, binding directly to the SR3G promoter in yeast. While *sr3g* and *wrky75* mutants showed contrasting salt stress responses, the double mutants of *sr3g/wrky75* displayed reduced sensitivity to salt stress.

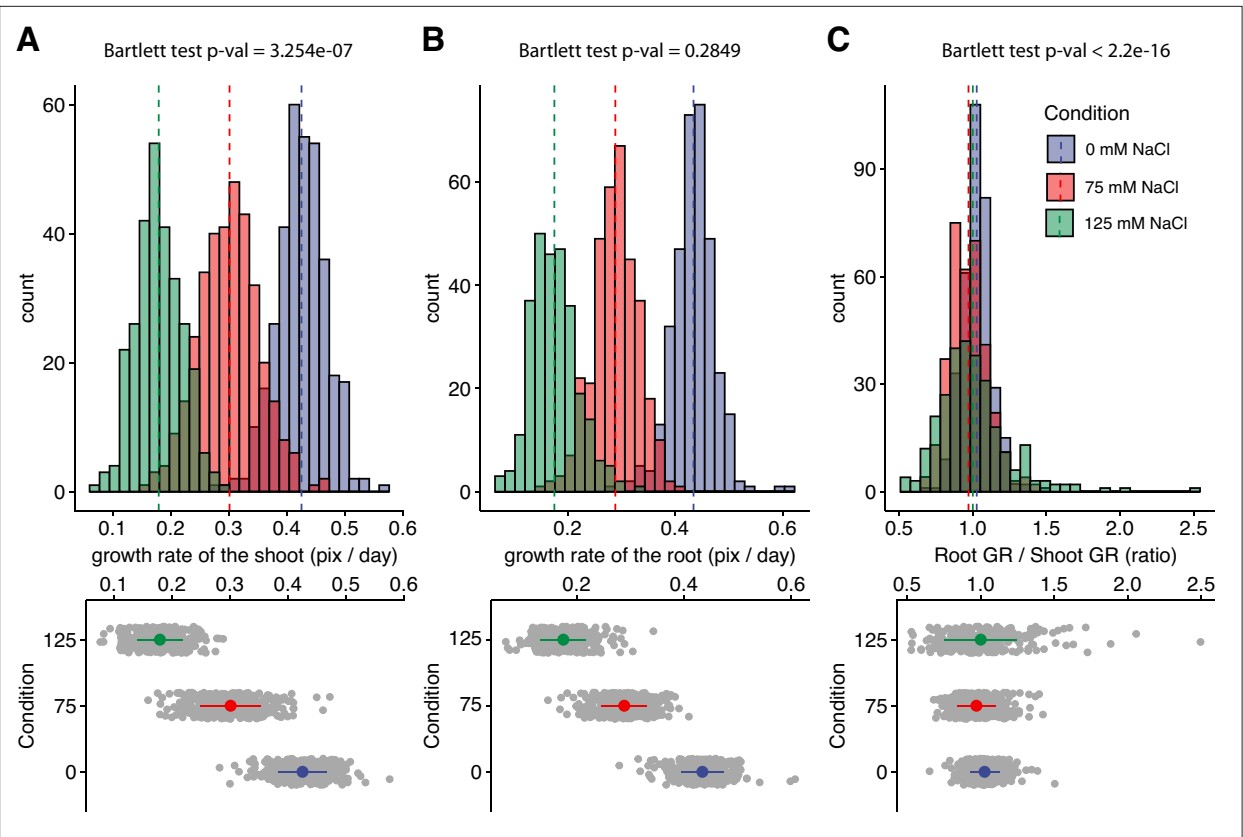

**Figure 1.** Salt stress is causing discoordination of root and shoot growth. *Arabidopsis* HapMap population was screened for salt stress-induced changes in root:shoot ratio. The increase in the projected area of shoot and root (*Figure 1—figure supplement 2*) was used to estimate (**A**) shoot and (**B**) root exponential growth rate. (**C**) The root:shoot growth rate ratio was calculated per genotype. The histograms represent the number of accessions across three studied conditions (0, 75, and 125 mM NaCl), whereas the population average is indicated using the dashed line. Additionally, the distribution of the genotypes within each treatment was visualized using the error plots (lower panel), where population average and standard error is indicated using a colored point and a line, respectively. Individual gray points represent individual genotypes.

The online version of this article includes the following figure supplement(s) for figure 1:

**Figure supplement 1.** Evaluation of the tool's precision for estimating seedling size.

**Figure supplement 2.** Salt stress reduced the increase in root and shoot area over time across HapMap accession.

**Figure supplement 3.** Salt stress induced changes in relative growth of shoot, root, and root:shoot ratio.

**Figure supplement 4.** The correlation between all measured traits and calculated stress tolerance indices (STI).

**Figure supplement 5.** The correlation between total seedling area and salt-induced changes in the root-to-shoot ratio.

**Figure supplement 6.** Region around NTL8 is associated with salt stress-induced changes in root-per-shoot growth.

**Figure supplement 7.** Region around unknown gene is associated with salt stress-induced changes in relative changes in shoot-per-total seedling area.

## Results

### Natural variation in salt stress-induced changes in root:shoot ratio

To investigate salt stress-induced changes in root:shoot ratio, we developed a custom tool that quantifies green and white pixels, corresponding to the shoot and root of *Arabidopsis* seedlings grown on agar plates (*Figure 1*). The projected area of root and shoot showed a high correlation with root and shoot fresh weight (*Figure 1—figure supplement 1*), indicating that the projected root and shoot area can be used as an accurate proxy to non-destructively measure the increase in root and shoot biomass from the images of seedlings grown on agar plates.

Using our custom tool, we re-analyzed the data consisting of 360 *Arabidopsis* accessions grown on plates supplemented with 0, 75, and 125 mM NaCl (*Julkowska et al., 2017*), for salt stress-induced changes in the root:shoot ratio (*Figure 1—figure supplement 2*). The growth of root and shoot for each plant was estimated using the exponential function. Salt stress-induced reduction in growth rate

was observed to be dose-dependent for both root and shoot (*Figure 1A and B*). In general, the root growth was more sensitive to salt stress compared to shoot growth, with the population-wide relative growth rates reduced to a fraction of 0.67 and 0.41 of the growth rates observed at control for 75 and 125 mM NaCl, respectively. In comparison, the growth rates of the shoot were significantly reduced to 0.71 and 0.43 of the control in 75 and 125 mM NaCl treatments, respectively (*Figure 1—figure supplement 3*). While the mean value of root:shoot growth rate did not change upon salt stress treatment, the variance in the root:shoot ratio significantly expanded with the increasing concentrations of salt (*Figure 1C*). These results suggest that while root and shoot growth are well coordinated under non-stress conditions, salt stress exposure results in loss of coordination of organ growth across *Arabidopsis* accessions. To test whether there is a clear directional correlation between the change in root:shoot ratio and overall salt stress tolerance, we have used the overall seedling size as a proxy for plant salt tolerance (*Figure 1—figure supplements 4–5*). No significant correlation was found between the root:shoot growth ratio and total seedling size (*Figure 1—figure supplements 4–5*), indicating that the relationship between coordination of root and shoot growth and salt tolerance during the early seedling establishment is complex.

## Identification of genetic components underlying salt-induced changes in root:shoot ratio

To identify genetic components underlying salt-induced changes in root:shoot ratio, we used the collected data as an input for GWAS. The associations were evaluated based on the p-value, the number of SNPs within the locus, and the number of traits associated with individual loci. As the Bonferroni threshold differs depending on the minor allele count (MAC) considered, we identified significant associations based on a Bonferroni threshold for each subpopulation of SNPs based on MAC (*Supplementary file 3*). While we conducted a GWAS on directly measured traits, as well as their Salt Tolerance Index (STI) values, however, the amount of associations with STI was much lower compared to directly measured traits (*Supplementary file 3*). This observation aligns with the understanding that plastic responses to environmental conditions tend to be genetically more complex. This complexity likely stems from the involvement of more genetic regulators compared to low-plasticity phenotypes.

The initial GWAS was performed using 250 k SNPs (*Horton et al., 2012*), and led to the identification of 14 significantly associated loci, of which 3 and 7 loci were associated with traits measured at 75 or 125 mM NaCl treatments, respectively. We identified one salt-specific association on chromosome 2 (*Figure 1—figure supplement 6*), where two SNPs were identified within AT2G27300 encoding NAC-domain containing transcription factor (NTL8). Additionally, salt-induced changes in root:shoot ratio and shoot:total seedling area were associated with SNPs on chromosome 3, where 2 significantly associated SNPs were flanking AT3G22430, encoding an unknown gene encoding a domain of unknown function (*Figure 1—figure supplement 7*).

To refine our analysis and further prioritize candidate genes for validation, we repeated the GWAS using a higher-density 4 M SNP panel (1001 *Genomes Consortium, 2016*). This increased SNP density provided more associations and highlighted loci with greater confidence due to the higher resolution. The higher-density panel also allowed us to capture different linkage disequilibrium (LD) blocks across the genome. One locus on chromosome 2, detected by both the 250 K and 4 M SNP panels, appeared to be in a region of strong LD or possibly under selection, consistently identified by both panels. However, most loci were not detected by the lower-density 250 K panel, likely due to differences in the panels' resolution and statistical power.

The new GWAS with the 4 M SNP panel revealed 32 additional loci, with only one locus (locus 30, *Supplementary file 3*). being detected by both the 250 K and 4 M SNP panels. We prioritized the identified loci based on several factors, including the number of significantly associated SNPs, MAC, and the number of traits associated with each locus. This led to the identification of three loci of major interest (*Supplementary file 4*). We further examined the regions in LD with the significantly associated SNPs for sequence divergence and open reading frames (*Figure 2—figure supplements 1–2*).

Out of identified associations, locus 30 stood out as it consisted of nine SNPs associated with p-value above the Bonferroni threshold, and was associated with both root:shoot and shoot:total seedling area at 6 and 8 d after transfer of the seedling to 125 mM NaCl (*Figure 2A, B, Figure 2— figure supplement 1, Supplementary file 4*). The significantly associated SNPs were located in and

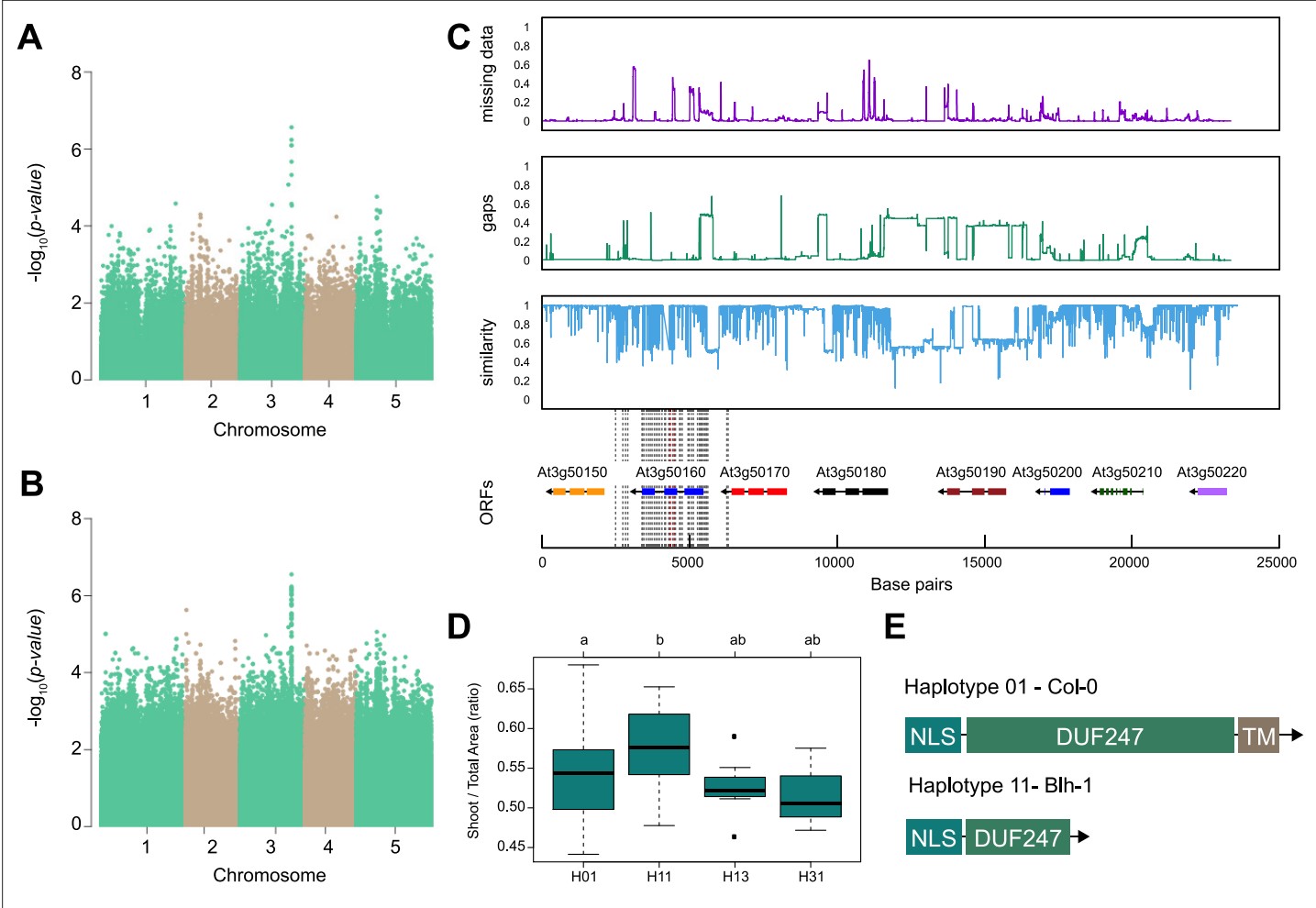

**Figure 2.** SR3G is associated with salt stress-induced changes in shoot-per-total seedling area ratio. The salt-induced changes in root-to-shoot and shoot-per-total seedling area ratios were used as an input for genome-wide association study (GWAS). Manhattan plots representing the associations of shoot-per-total seedling area recorded 6 d after transfer to 125 mM NaCl and (**A**) 250 k SNPs and (**B**) 4 M SNPs. (**C**) Significant associations were found with the SNPs forming a locus on chromosome 3 in and around *SR3G* (AT3G50160), encoding Domain of Unknown Function 247 (DUF247). The natural variation in the linkage-disequilibrium (LD) region was studied in 162 accessions sequenced by 1001 Genomes Project. The upper panel represents portion of the missing data, upper middle panel represents deletions relative to Col-0, while lower middle panel represents the sequence similarity compared to Col-0. The bottom panel represents the open reading frames (ORFs) within the LD, and the location of associated SNPs is indicated with the dashed lines. Red dashed lines represent associations above Bonferroni threshold in 250 k SNP mapping, while gray lines represent associations with -log10(p-value)>5 in the 4 M SNP mapping. (**D**) The haplotype analysis performed using SNPs located within the coding region of the SR3G revealed significant differences in shoot-per-total seedling area recorded 6 d after transfer to 125 mM NaCl between Haplotype groups 1 (represented by 46 accessions, including Col-0) and 11 (represented by 11 accessions, including Blh-1). The significant differences between individual haplotype groups were tested using ANOVA with Tukey HSD to identify significantly different groups. (**E**) Upon further sequencing of SR3G in accessions from Haplotype 1 and 11, revealed two 200 bp insertions within SR3G exons in four tested accessions belonging to haplotype 11 group. These 200 bp insertions resulted in a premature STOP-codon within the DUF247 domain (after Gly-215).

The online version of this article includes the following figure supplement(s) for figure 2:

**Figure supplement 1.** The QQ-plots for genome-wide association study (GWAS) models with shoot-per-total seedling area.

**Figure supplement 2.** Haplotype analysis of AT3G50160 (SR3G).

**Figure supplement 3.** Nucleotide sequence alignment of SR3G CDS between Col-0 and Blh-1 alleles.

**Figure supplement 4.** Amino acid sequence alignment of SR3G CDS between Col-0 and Blh-1 alleles.

**Figure supplement 5.** Sequence alignment between Col-0 and Blh-1 alleles for SR3G promotor.

around the gene coding region of AT3G50160, which encodes a domain of unknown function 247 (DUF247). We investigated the region for sequence divergence, and found that the region contained not only a high degree of sequence diversity but also potential structural variants (*Figure 2C*). To identify allelic variation underlying the initial associations, we performed haplotype analysis, where accessions with shared SNP patterns within the protein-coding part of AT3G50160 were grouped into individual haplotype-groups (*Figure 2D*). Each unique haplotype, represented by at least three accessions, was subsequently studied for differences in observed phenotypes. We observed that Haplotype 1, represented by 46 accessions including Col-0, was significantly different from Haplotype 11, represented by 11 accessions including Blh-1, in shoot-per-total seedling area at 6 d after transfer to 125 mM NaCl (*Figure 2D*), as well as shoot growth rate and relative shoot growth rate at 125 mM NaCl (*Figure 2—figure supplement 2*). Upon further investigation and sequencing of AT3G50160 in both Col-0 and Blh-1 accessions, we discovered that the Blh-1 allele carries a large insertion at 644 nucleotides downstream from the ATG of AT3G50160, followed by multiple other sequence variations (*Figure 2—figure supplement 3*). The first insertion leads to a premature STOP-codon within the DUF247 domain after Gly-215 (*Figure 2E*, *Figure 2—figure supplement 4*), resulting in a truncated protein that still carries a nuclear localization signal and coiled-coil domain, but is missing a major part of the calcium-binding domain and a predicted transmembrane domain (*Figure 2—figure supplement 5*). As there are other DUF247s within the locus associated with salt-induced changes in root:shoot ratio, we decided to name the gene containing the majority of the associated SNPs (At3g50160) as Salt Root:shoot Ratio Regulator Gene (SR3G). Together, these results suggest that the allelic variation in the SR3G protein sequence might underlie the observed differences in root:shoot ratio under salt stress conditions.

## Evolutionary context of the SR3G locus reveals its origin in local genome duplication

SR3G sits in what appears to be a tandemly duplicated array of DUF247s along *Arabidopsis* chromosome 3. The genome-wide identification of SR3G included eight DUF247 paralogs, AT3G50120, AT3G50130, AT3G50140, AT3G50150, AT3G50170, AT3G50180, AT3G50190, and AT3G50200. Reciprocal best BLAST within CoGe's BLAST tool of these nine DUF247 paralogs against the genomes of seven other Brassicaceae (see *Figure 3A*, *Supplementary file 5*) revealed a total of 50 potential orthologs used for downstream analysis. While the copy number of DUF247 loci are similar across the examined species, the noted exceptions are those with recent whole genome duplication events (*Brassica rapa* and *Camelina sativa*). Given these data, it appears that DUF247 loci are commonly tandemly duplicated, although the eight paralogs observed in *Arabidopsis* represents an extreme case for this gene family.

To examine conservation within the predicted DUF247 functional domains, we generated an amino-acid based-multiple sequence alignment using MUSCLE. All of the identified DUF247 proteins contained the DUF247 domain near the center of the translated sequence. While partial deletions were identified outside of the DUF247 domain at the C-terminus of a few DUF247s, such as *CARHR159180*, *Csa09g052970*, and *Bra01902*, in general, the DUF247 domains were highly conserved across Brassicaceae. The amino acid sequences spanning the 'coiled-coil domain' (Amino acid position: Met153-Lys173), putative $Ca^{2+}$ binding domain (Amino acid position: Gly210-Phe372), and super polar fragment (Amino acid position: Ser301-Ile320) were further selected for phylogenetic reconstruction (*Figure 3B*). The resulting phylogeny demonstrated that SR3G is sister to a clade containing AT3G50150 and representatives from *C. sativa*, *Capsellarubella,* and *A. lyrata*, with representative DUF247s from *B. rapa* and *Eutremasalsugineum* serving as a monophyletic outgroup. These data support a hypothesis where SR3G is a recent duplicate (likely A/C Clade specific; *Forsythe et al., 2020*) of AT3G50150 and that representative SR3G sequences in *C. sativa*, *C. rubella*, and *A. lyrata* have been lost.

Given the specific retention of AT3G50150 in *A. thaliana*, we next tested whether this locus is experiencing directional selection. To test this assumption, we grouped the branches leading to SR3G into four groups (*Figure 3B*: $\omega 0$, $\omega 1$, $\omega 2$, and $\omega$-rest) and then performed a branch-site model A hypothesis test. In our null hypothesis ($H_0$), we hypothesized that all branches have the same selection pressure ($H_0$: $\omega$-rest branches = $\omega 0 = \omega 3 = \omega 1 = \omega 2$; see *Supplementary file 5* for omega values for each hypothesis). For our alternative hypothesis, we created five different $\omega$ groups to cover

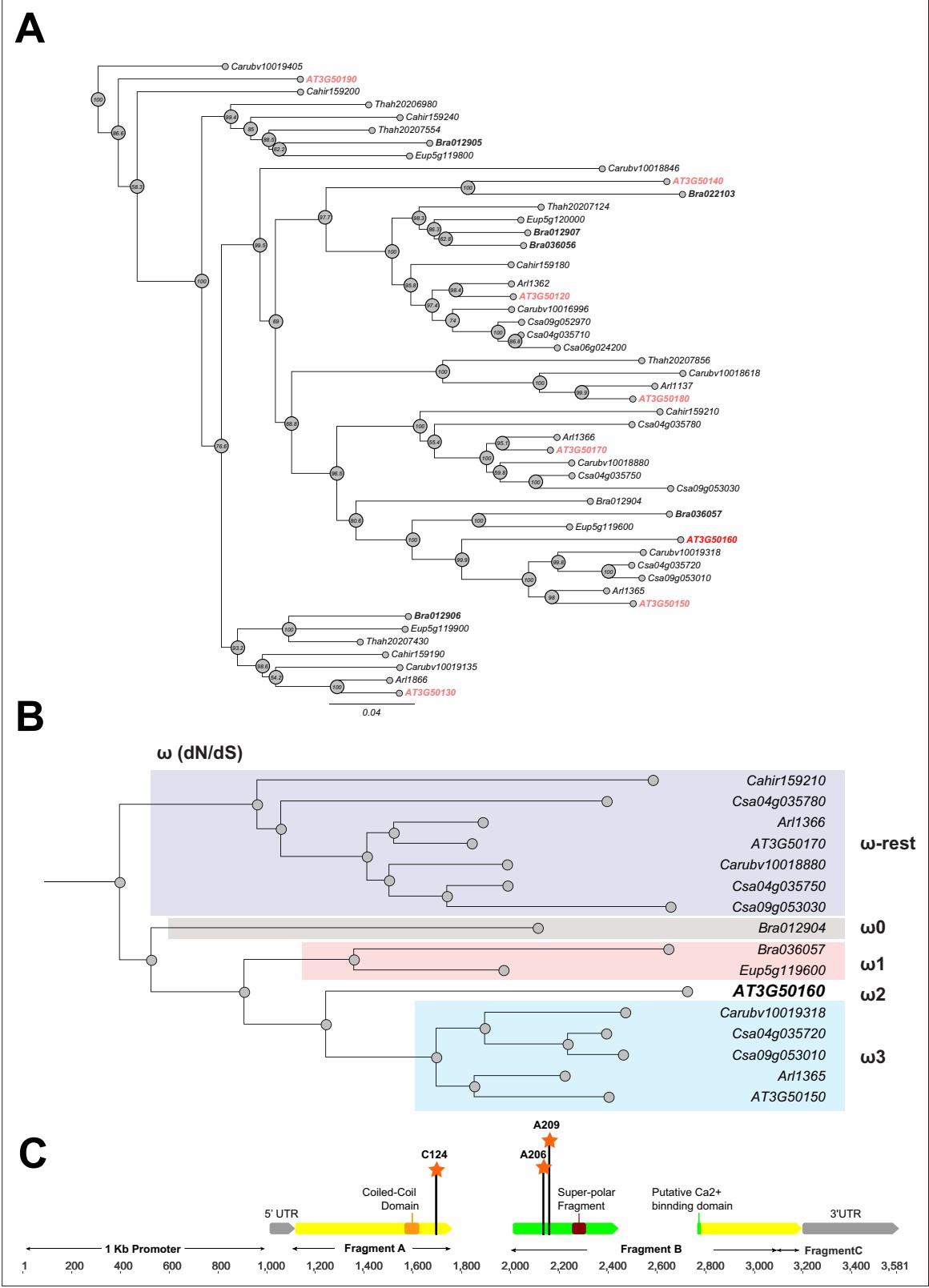

**Figure 3.** Phylogeny and positive selection analysis of *SR3G* and its orthologs from other species. The orthologous genes of *SR3G* (AT3G50160) from other seven species were identified using GoGeBlast (E-value <1.00E-10). (**A**) The protein sequences of the 50 homologous genes were aligned by MUSCLE and then an unrooted phylogenetic tree was reconstructed by a Maximum-likelihood (ML)-derived style by RAxML (Bootstrap number: 100). (**B**) A positive selection analysis was conducted by branch-site model A test: Specifically, we hypothesized the differential or constant dN/dS ($\omega$)

*Figure 3 continued on next page*

*Figure 3 continued*

substitution rate among the closely distant branches leading to DUF247 (AT3G50160) (See Method and materials for details). These branches were marked as ω1 (light brown branch), ω2 (DUF247-specific branch), ω3 (light blue branch), and ω-rest (light purple branch). (**C**) The gene structure of DUF247 (AT3G50160) was illustrated by wide arrows (gray arrow: 5 and 3' UTR, yellow arrow: coding-region). The regions used for cloning (Fragments A, B, and C) were marked by narrow arrows along with 'Coil-Coil Domain' and 'Super-polar Fragment' marked on Fragments A and B, respectively. Lastly, the Bayes Empirical Bayes (BEB) was performed to test the probability of sites along with ω>1 over the DUF247 (AT3G50160). Three sites were generated with a posterior probability >0.90: 124 C (p=0.992), 206 A (p=0.951), and 209 A (p=0.988). These three sites were marked with an orange asterisk and assumed to have been undergoing positive selection.

scenarios where specific branches may be experiencing directional selection (See Methods). Based on our analysis, the branch leading to SR3G is experiencing directional selection (likelihood score = 20625.5,, Chi-square derived P value; p=0.00418). Interestingly, no other branches appeared to be experiencing the same selection. To pinpoint specific amino acids within SR3G under selection, we performed the Bayes Empirical Bayes (BEB) test, identifying three sites with a posterior probability >0.90, including 124 C (p=0.992), 206 A (p=0.951), and 209 A (p=0.988). Two out of three sites are located at the putative $Ca^{2+}$ binding site and all three sites possess a unique amino acid compared with the other 49 amino acid sequences (***Figure 3C***).

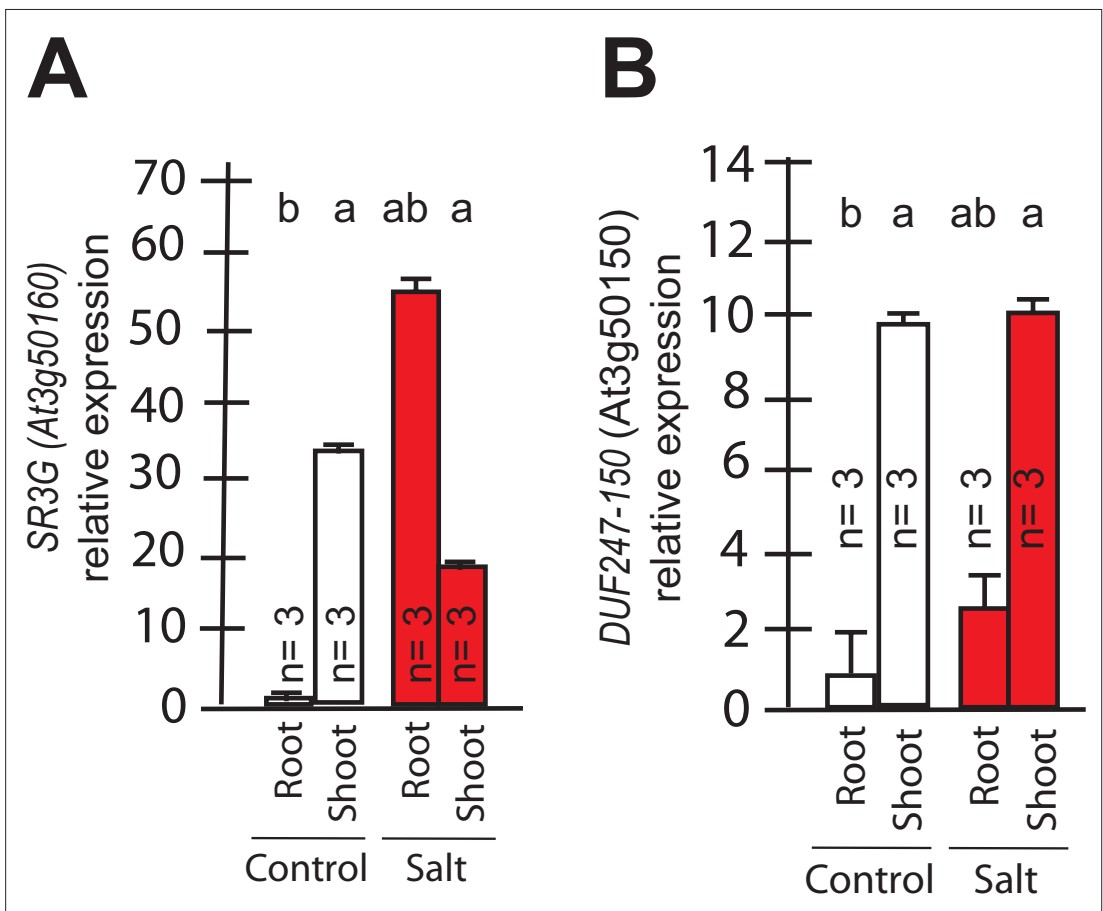

**Figure 4.** Transcript abundances for the two closely related *DUF247s*. Expression of (**A**) *SR3G* (AT3G50160) (**B**) and its closest homolog *DUF247-150* (AT3G50150) were measured in Col-0 seedlings grown with and without salt stress. RT-qPCR analyses were conducted using seedlings grown on 1/2 x MS for 4 d and then followed by transferring to the treatment plates with or without 75 mM NaCl for 1 wk. Mean values are shown ±SE, with number of replicates (n) shown in each graph. *AT4G04120* was used as a reference gene for normalization. Significance was determined by the Tukey–Kramer HSD test in JMP.

The online version of this article includes the following figure supplement(s) for figure 4:

**Figure supplement 1.** Gene expression profile of *SR3G* across various developmental stages and salt stress.

**Figure supplement 2.** Gene expression profile of other *DUF247s* under salt stress.

## SR3G expression is enhanced in the root by salt stress

To identify the developmental window and specific plant tissues where the *SR3G* gene is active, we examined its expression pattern. Based on *Arabidopsis* eFP Browser (http://bar.utoronto.ca/), *SR3G* expression is relatively low, and restricted to the early stages of seedling development (*Figure 4A*, *Figure 4—figure supplement 1A*). Upon examining the expression profiles specific to different tissues, we observed that *SR3G* is predominantly expressed in the root stele (*Figure 4—figure supplement 1B*). Moreover, *SR3G* expression was induced by salt stress (*Figure 4—figure supplement 1C*). To validate these results, we performed RT-qPCR on 11- d-old seedlings of Col-0 exposed to 0 or 75 mM NaCl for 7 d. In Col-0 seedlings, the transcript levels of *SR3G* were significantly higher in shoot compared to the root tissue (*Figure 4A*). Interestingly, the *SR3G* transcripts increased only in root upon salt stress exposure but not in shoot (*Figure 4A*). These results suggest that *SR3G* is the most prominent DUF247 studied within our study to affect root development under salt stress.

We further examined expression profiles of other *DUF247* genes that directly neighbor *SR3G* in response to salt stress using the eFP Browser. We observed that the expression of DUF247-120 (At3g50120) remained undetectable under control conditions, except for a slight increase in rosette leaves after 6 hr of salt stress exposure (*Figure 4—figure supplement 2*). For DUF247-130 (At3g50130), –140 (At3g50140), –150 (At3g50150), and –170 (At3g50170), high expression was detected in roots under non-stress conditions, but expression levels decreased upon salt stress (*Figure 4—figure supplement 2*). The expression of DUF247-180 (At3g50180) remained unchanged with or without salt stress (*Figure 4—figure supplement 2*). DUF247-190 (At3g50190) exhibited high expression in roots during early salt stress exposure, which then declined at later stages (*Figure 4—figure supplement 2*). For DUF247-200 (At3g50200), its already low detectable expression disappeared entirely during prolonged salt stress exposure (*Figure 4—figure supplement 2*). Because *DUF247-150* (At3g50150) is closely located to the *SR3G* in chromosome 3, we investigated its gene expression in young seedlings. We observed that its expression was significantly higher in the shoot compared to the root tissue. However, its expression remained unchanged in both root and shoot under salt stress (*Figure 4B*). Together, these results suggest that *SR3G* is the most prominent DUF247 studied herein involve in regulation of root development under salt stress.

## SR3G relocates from plasma membrane to nucleus/cytosol after removal of its transmembrane domain

SR3G protein is predicted to contain a nuclear localization signal (Lys15-Ala19), a coiled-coil domain (Met153-Lys173), a $Ca^{2+}$ binding pocket (Gly210-Phe372), and a trans-membrane domain (Pro469-Thr503) (*Figure 5—figure supplement 1*). We generated constitutive expression constructs of full-length *SR3G* gene CDS, and a combination of individual domains (refer to as A, B, and AB domains) to identify their subcellular localization (*Figure 5—figure supplement 1*, *Supplementary file 6*). Fragment A comprised a nuclear localization signal, a coiled-coil domain, and the initial segment of the $Ca^{2+}$ binding pocket, encompassing seven amino acids. Fragment B encompassed the remaining segment of the $Ca^{2+}$ binding pocket. Fragment C only included the transmembrane domain. All constructs contained N-terminal fusion of mVenus. The individual constructs were used in transient expression assay in tobacco leaves. Our result showed that full-length SR3G localizes to the plasma membrane (*Figure 5A*) as its signal overlapped with FM4-64, the plasma membrane marker (*Figure 5B*). The SR3G fragments from which the transmembrane domain was excluded— namely fragments A, B, and AB—exhibited localization in both the nucleus and cytosol, resembling the pattern observed with free-GFP localization (*Figure 5C–F*). It is noteworthy that while the signal from fragment A was directed to both the cytosol and nucleus, its intensity was notably stronger in the nucleus (*Figure 5C*), aligning with the prediction that it contains the nuclear localization signal. Together, these findings indicate that the positioning of SR3G within the plasma membrane is due to the transmembrane domain encoded within the fragment C, and its removal could potentially release the protein into both the nucleus and cytosol.

## Lower expression of *SR3G* results in reduced lateral root development under salt stress

To investigate the role of *SR3G* in salt-induced changes in root growth and development, we studied the root system architecture of two T-DNA insertion lines: *sr3g-4* (SAIL_690_E12) and *sr3g-5*

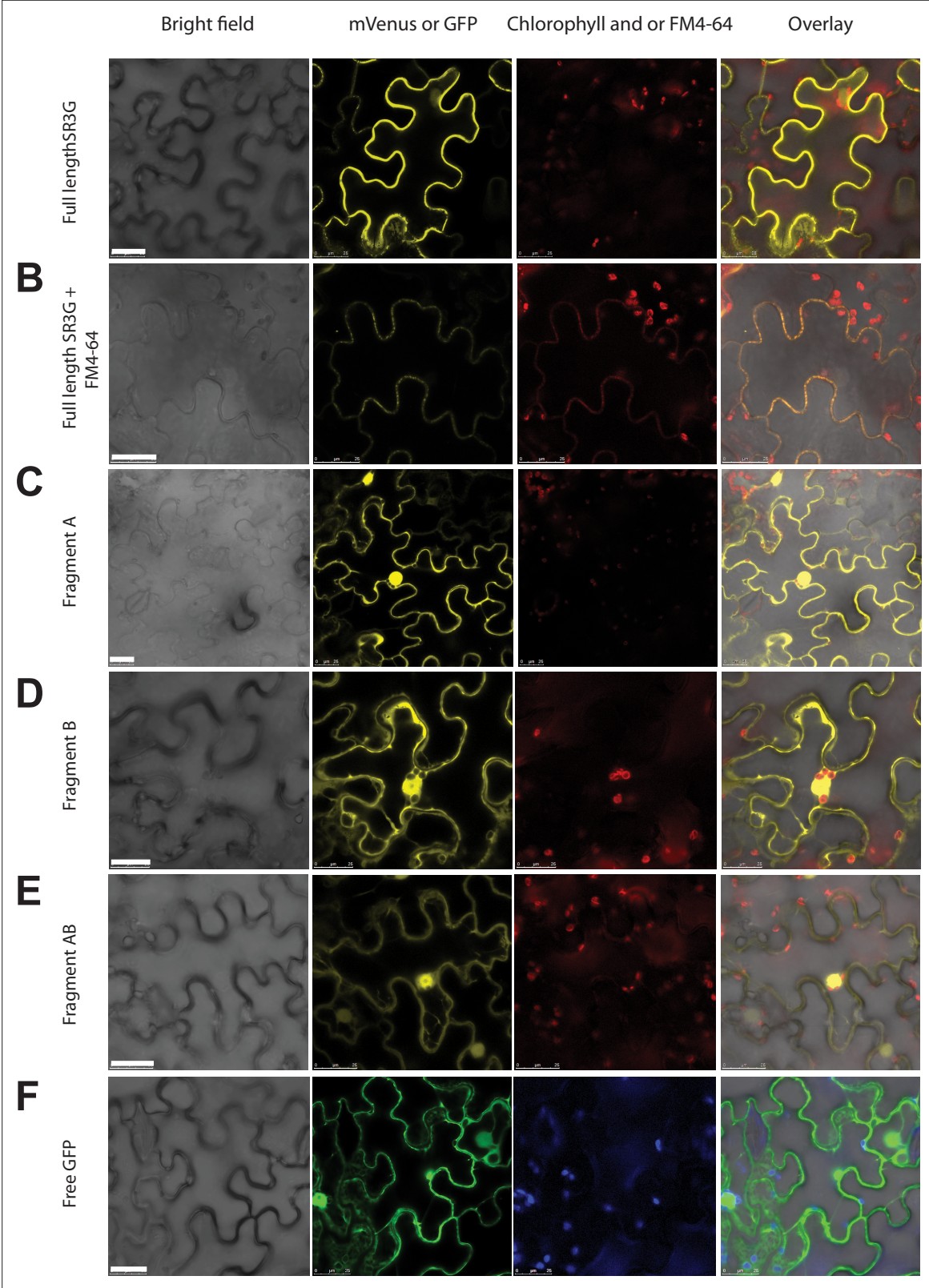

**Figure 5.** SR3G full-length protein resides in the plasma membrane while removal of its transmembrane domain results in protein relocation to nucleus and cytosol. (**A-B**) SR3G full-length protein or (**C-E**) truncated versions fused to mVenus at the N-terminus were agro-infiltrated into the *Nicotiana benthamiana* leaves for transient expression. Shown are a bright-field image of transfected leaf cells and mVenus-SR3G-mediated fluorescence as well as GFP-based subcellular marker. (**A**) Localization of full-length SR3G alone or (**B**) in combination with the known plasma membrane dye, FM4-64.

*Figure 5 continued on next page*

*Figure 5 continued*

Localization of truncated versions of SR3G are shown for (**C**) SR3G-Fragment A, (**D**) SR3G-Fragment B, and (**E**) SR3G-Fragment AB. (**F**) Free GFP was used as a nuclear and cytosolic marker. Scale bar = 25 µm.

The online version of this article includes the following figure supplement(s) for figure 5:

**Figure supplement 1.** SR3G predicted protein domains.

(SAIL_608_C06) localized in second exon and first intron region of the *SR3G* gene, respectively (*Figure 6—figure supplement 1A*). Among these two mutant lines, *SR3G* expression was reduced in the *sr3g-5* mutant compared to Col-0 (*Figure 6—figure supplement 1B*), indicating the *sr3g-5* is a knockdown mutant. We did not observe any differences in *SR3G* transcript abundance between *sr3g-4* and Col-0 (*Figure 6—figure supplement 1B*). Moreover, we examined the transcript abundance of the closely related gene (i.e. *DUF247-150, At3g50150*) in the *sr3g-5* mutant, and found no significant difference. In the case that *DUF247-150* would act redundantly to *SR3G*, we would expect that the expression of *DUF247-150* would be increased in *SR3G* knockdown mutant. As this is not

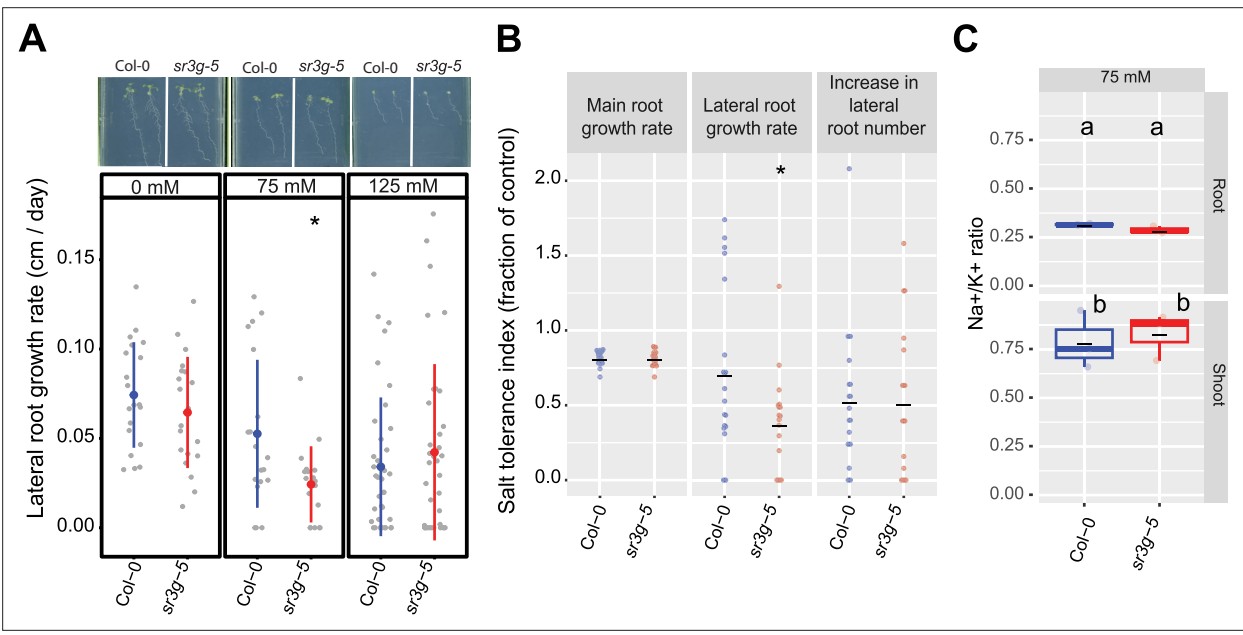

**Figure 6.** *sr3g* mutant displays reduced growth rate for the lateral root length. Root system architecture analysis of Col-0 and *sr3g-5* plants under various concentrations of NaCl are shown here. (**A**) The representative images of 13-d-old Col-0 and *sr3g-5* genotypes that experienced 9 d of salt treatment at indicated concentrations as well as growth rate for lateral root. (**B**) Salt Tolerance Index (STI) for the main root length, average lateral root length, and lateral root number at 75 mM NaCl. The STI was calculated by dividing the growth rate measured under salt stress by the growth rate measured under control condition for each genotype. (**C**) Na⁺/K⁺ ratio in root and shoot of Col-0 and *sr3g-5* after 2 wk on 75 mM salt are shown. Each dot in (**A**) and (**B**) represents individual replicate per genotype. Lines in (**A**) and (**B**) graphs represent median and average, respectively. In (**A**) and (**B**), the asterisks above the graphs indicate significant differences between Col-0 and the *sr3g-5* mutant, as determined by the Student 's t-test: *p<0.05. Statistical analysis in (**C**) was done by comparison of the means for all pairs using Tukey–Kramer HSD test. Levels not connected by the same letter are significantly different (p<0.05). Root system architecture and ICP-AES analyses of Col-0 and *sr3g-5* mutant are shown in details in *Figure 6—figure supplement 2*.

The online version of this article includes the following figure supplement(s) for figure 6:

**Figure supplement 1.** The expression of *DUF247-150* remains unaltered in the *sr3g* mutant.

**Figure supplement 2.** *sr3g-5* mutant displays no difference in main root length and lateral root numbers compared to Col-0.

**Figure supplement 3.** *sr3g-4* mutant has no alteration in root system architectures compared to Col-0.

**Figure supplement 4.** Root system architecture analysis of neighboring DUF247 mutants under the salt.

**Figure supplement 5.** Root system architecture analysis of neighboring DUF247 mutants under the salt, continues.

**Figure supplement 6.** Root system architecture analysis of RNAi lines targeting neighboring DUF247s.

**Figure supplement 7.** Root system architecture analysis of RNAi lines targeting neighboring DUF247s continued.

**Figure supplement 8.** Root system architecture analysis of RNAi lines targeting neighboring DUF247s continued.

the case, these results further contribute to the evidence that *SR3G* and *DUF247-150* are not acting redundantly (*Figure 6—figure supplement 1C*).

Two T-DNA insertion lines were subsequently studied for salt-induced alterations in root architecture (*Figure 6*, *Figure 6—figure supplement 2*, *Figure 6—figure supplement 3*). Among the various features of the root system architecture that we investigated (*Figure 6—figure supplement 2*), *sr3g-5* exhibited a significantly reduced growth rate of lateral roots under 75 mM NaCl, but not under 125 mM NaCl, when compared to Col-0 (*Figure 6A*). Likewise, the *sr3g-5* mutant displayed a significant reduction in the Stress Tolerance Index (STI), calculated as the ratio of growth rate under salt to growth rate under control conditions, for lateral root length under 75 mM NaCl (*Figure 6B*). No significant difference was observed between *sr3g-4* and Col-0 (*Figure 6—figure supplement 3*), except for STI calculated using growth rates of lateral root length under 125 mM salt stress (*Figure 6—figure supplement 3E*). In this case, the sr3g-4 mutant exhibited a non-significant decrease compared to Col-0, consistent with the observations made for the *sr3g-5* mutant line (*Figure 6B*). Based on the RT-qPCR results (*Figure 6—figure supplement 1*), we proceeded to further investigate the role of *SR3G* in salt stress, focusing on the sr3g-5 mutant in our subsequent investigations.

As $Na^+$ exclusion from the shoot is one of the main mechanisms of salt tolerance (*Julkowska et al., 2017*; *Møller et al., 2009*; *Munns and Tester, 2008*), we further investigated the role of *SR3G* in ion exclusion using plate-grown seedlings. We used root and shoot samples collected for Col-0 and *sr3g-5* mutant to examine $Na^+$ and $K^+$ accumulation using ICP-AES at 14 d after exposure to salt stress. We observed no significant differences between Col-0 and *sr3g-5* mutant for $Na^+$ and $K^+$ accumulations and their corresponding ratio upon salt stress exposure (*Figure 6C*, *Figure 6—figure supplement 2*). These results suggest that *SR3G* plays a role in root development under the salt, but does not affect ion accumulation at early developmental stages.

To evaluate the effects of other closely related DUF247 genes on root development, we have investigated the T-DNA insertion lines targeting the neighboring DUF247 genes (AT3G50120, AT3G50130, AT3G50140, AT3G50150, AT3G50170, AT3G50180, AT3G50190, AT3G50200), and RNAi lines targeting one or more DUF247s within the interval (*Supplementary file 4*, *Figure 6—figure supplements 4–8*). For the screened T-DNA insertion lines (*Figure 6—figure supplements 4–5*), we have not observed consistent changes in main or lateral root development in either control or salt stress conditions across alleles targeting the same gene. The only exception was *DUF247-200* encoded by AT3G50200, where two out of three studied T-DNA insertion alleles exhibited an increased salt tolerance index for main root growth rate (*Figure 6—figure supplement 5*).

For the screened RNAi lines (*Supplementary file 4*, *Figure 6—figure supplements 6–8*), we observed the most interesting root architecture phenotypes in lines targeting the neighboring genes of the *SR3G* (RNAi line #04, targeting AT3G50150, RNAi line #16 targeting AT3G50130, *SR3G*, and AT3G50170, and RNAi line #19 targeting AT3G50130, AT3G50140, AT3G50150, *SR3G*, AT3G50170, AT3G50180, and AT3G50190). Three out of five screened independent transformation lines from RNAi line #04 showed a decreased relative main root length (Salt/Control, CSHL #4 B2a, D4a, and G1d), whereas two lines showed a decrease in relative lateral root number (CSHL #4 D4a and F1b), and one line showed decreased lateral root length at 75 mM NaCl (CSHL #4 D4a, *Figure 6—figure supplements 6–8*). Two out of two independent transformants for RNAi line #16 showed a decreased relative main root length and increased lateral root number at 75 mM NaCl (CSHL #16 A2c and B1a, *Figure 6—figure supplements 6–8*). In three of the four screened independent transformants of line #19 we observed an increased lateral root length at 75 mM NaCl (CSHL #19 F1a, I3c, and L1c, *Figure 6—figure supplements 6–8*). We did not observe strong reduction in lateral root development across RNAi lines targeting two closely related DUF247 genes (i.e. RNAi line #22, targeting AT3G50150, and *SR3G*), as in the single mutant (*Figure 6*, *Figure 6—figure supplement 6*). The above results further support that these two genes do not act redundantly in the context of lateral root development.

## Loss of *SR3G* results in increased salt tolerance in soil

To investigate *SR3G's* contribution to salt stress tolerance beyond seedling stage and agar plate conditions, we evaluated the performance of Col-0 and *sr3g-5* mutant in soil. We exposed the soil-grown plants to salt stress and imaged the rosette size every 30 min for 2 wk using the automated imaging platform (*Yu et al., 2023*). As salt-induced changes depend on the developmental stage at

which the stress is applied (*Julkowska et al., 2017*), we examined the role of *SR3G* exposed to salt stress at 2 or 3 wk after germination, corresponding to early and late vegetative stages. We calculated a daily growth rate for each plant, using the projected rosette area. No significant difference was observed between Col-0 and *sr3g-5* under control conditions (*Figure 7A and B*), indicating that *SR3G* does not play a role in rosette growth and development under non-stress conditions. Significant difference was observed between Col-0 and *sr3g-5* for plants exposed to early salt treatment (*Figure 7A and B*), where *sr3g-5* mutant displayed an increased rosette growth rate as compared to Col-0 (*Figure 7A*). Late stress exposure resulted in significantly higher *sr3g-5* growth rate at 2 and 3 d after exposure to salt stress (*Figure 7A*). Similarly, the *sr3g-5* mutant showed a significantly higher rosette area compared to Col-0 exclusively when plants were exposed to salt stress at an earlier stage of their development (*Figure 7B*). No significant difference was observed between Col-0 and *sr3g-5* mutant in terms of water content under either of conditions tested, providing no substantial evidence for SR3G to regulate water content (*Figure 7—figure supplement 1*).

To investigate the role of *SR3G* in the maintenance of cellular integrity during salt stress, we evaluated the salt-induced changes in cell membrane integrity by measuring ion leakage in the leaf discs at the end of the experiment (*Figure 7C*). While ion leakage increased in Col-0 under early salt stress conditions, no noticeable salt-induced change was observed for *sr3g-5* compared to control condition (*Figure 7C*). Although the *sr3g-5* mutant exhibited significant increase in ion leakage in response to late salt stress, this phenomenon may be more closely associated with the onset of senescence rather than direct exposure to salt stress (*Figure 7C*).

We further explored the role of *SR3G* in ion exclusion by quantifying the accumulation of $Na^+$ and $K^+$ ions in the rosette at the end of the experiment (4-wk-old plants). We observed a reduced accumulation of $Na^+$ in *sr3g-5* mutants compared to Col-0 for plants exposed to salt stress at an earlier time-point (*Figure 7D*). No significant differences in $Na^+$ accumulation was observed between the Col-0 and *sr3g-5* at the late salt stress regime (*Figure 7D*). A subtle, non-significant, reduction in $K^+$ content was observed in *sr3g-5* mutant compared to Col-0 across all three conditions studied (*Figure 7E*). In summary, these results suggest that *SR3G* plays a negative role in maintenance of plant growth, cellular integrity and $Na^+$ accumulation during the salt stress, especially when the salt stress is applied during early vegetative development.

## *sr3g* mutant displays enhanced root suberization

As *SR3G* limits shoot growth, promotes cell damage and sodium translocation into shoot (*Figure 7*), and is expressed in the most inner layers of the root (i.e. stele, *Figure 4—figure supplement 1*), we hypothesized that *SR3G* plays a role in the regulation of suberization within the Casparian strips located in the endodermis, the layer adjacent to the stele. Salt stress exposure increases root suberization, thereby limiting $Na^+$ loading into the transpiration stream and accumulation in the shoot (*Baxter et al., 2009*). We examined root suberization patterns in Col-0 and *sr3g-5* under non-stress and salt stress conditions. Root suberization was visualized using lipophilic stain Fluorol Yellow (FY), and staining intensity was examined under a confocal microscope. Additionally, we examined the developmental transition in root suberization through imaging four regions along the main root (*Figure 8*). Visually, we observed that roots of *sr3g-5* were more suberized compared to Col-0 under both control and salt stress conditions (*Figure 8A*). The quantification of FY signal intensity revealed non-significant, but substantially higher levels of root suberization in *sr3g-5* compared to Col-0 in sections one to three of the root under control condition (*Figure 8B*). Only in root section four, the level of suberization significantly decreased in *sr3g-5* compared to Col-0 (*Figure 8B*). In plants exposed to salt stress, we observed a non-significant difference between *sr3g-5* and Col-0 throughout the four sections, although, in section one which is the root tip proximal, *sr3g-5* showed more suberization compared to Col-0 (section 1, *Figure 8A*). These results suggest that *SR3G* may play a negative role in root suberization.

To validate the FY histochemical analysis, we quantified root suberin monomers in the *Arabidopsis* seedlings using GC-MS analysis. Suberin monomers that were detected in the *Arabidopsis* roots included fatty acid (FA), fatty alcohol (OH-FA), α, ω -dicarboxylic acids (DCA), and ω -hydroxy fatty acid ( ω OH-FA). Roots of *sr3g-5* contained significantly more total FA and DCA monomers compared to Col-0 under control condition, whereas the total OH-FA and ω OH-FA remained unchanged (*Figure 8C*). In plants exposed to salt stress, the *sr3g-5* mutant showed an increased abundance of

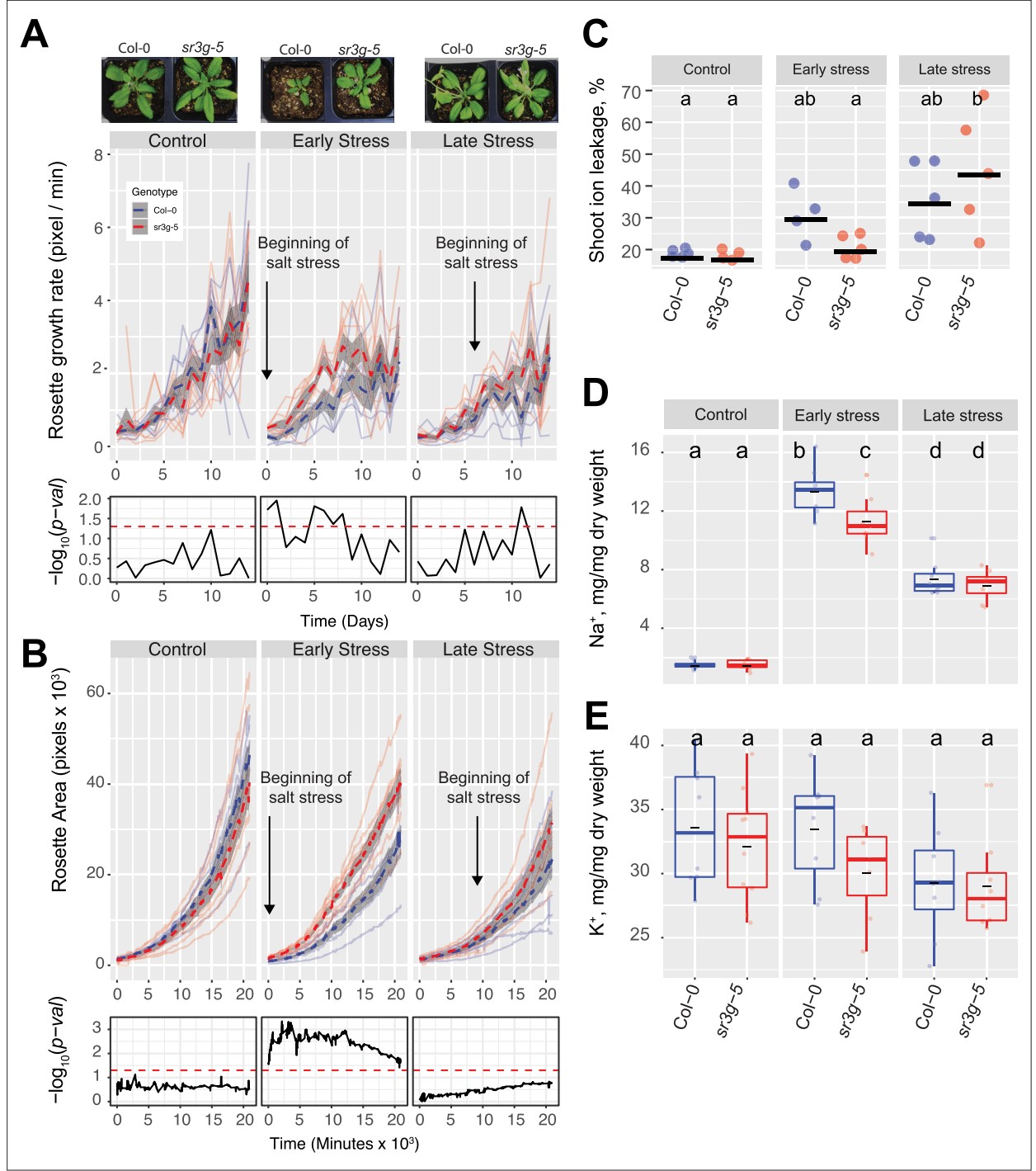

**Figure 7.** *sr3g* mutant has higher growth rate and larger rosette area while displaying less ion leakage and Na$^+$ accumulation in shoot under salt stress. The salt stress responses of 2-wk-old (referred to as 'early salt stress') or 3-wk-old (referred to as 'late salt stress') soil-grown Col-0 and *sr3g-5* plants that were exposed to a final concentration of 100 mM NaCl were examined here. (**A**) Rosette growth rate and (**B**) area were monitored over a period of 2 wk, during which the 'early salt stress' and 'late salt stress' groups were exposed to 2 or 1 wk of salt stress, respectively. Each line represents the trajectory of individual plant throughout time of experiments, where red and blue lines indicate Col-0 and *sr3g-5* plants, respectively. The dashed lines represent the mean values of the genotype per condition, whereas the gray band represents the confidence interval. The significant differences between genotypes, determined by one-way analysis of variance, are illustrated in a plot below each graph, with dashed red line representing a threshold corresponding to p-value of 0.05. Representative images of 4-wk-old Col-0 and *sr3g-5* plants were shown in (**A**). (**C**) Ion leakage percentage, (**D**) Na$^+$ and (**E**) K$^+$ contents in shoot were measured in 4-wk-old plants. The asterisks above the graphs indicate significant differences between Col-0 and *sr3g-5* plants, as determined

*Figure 7 continued on next page*

**Figure 7 continued**

by one-way analysis of variance: *p<0.05, **p<0.01, ***p<0.001, and ****p<0.0001. Statistically different groups were determined using Tukey–Kramer HSD test. Groups that are not assigned the same letter are significantly different (p<0.05).

The online version of this article includes the following figure supplement(s) for figure 7:

**Figure supplement 1.** Additional phenotypes of soil grown sr3g mutant and Col-0 wild-type plants.

three suberin monomers, with DCA and $\omega$OH-FA showing the most significant changes (*Figure 8C*). The total FA decreased significantly in the mutant compared to Col-0 (*Figure 8C*). Together, these results further corroborate with FY staining, suggesting that SR3G plays a negative role in root suberization.

## *SR3G* expression is regulated by *WRKY75* transcription factor

To identify potential transcriptional regulators of *SR3G*, we examined the *SR3G* promoter for cis-regulatory elements using the CisCross tool (https://plamorph.sysbio.ru/ciscross/), and made a list of transcription factors that putatively interact with *SR3G* promoter (*Figure 9A*). As the root stele showed the highest enrichment in the *SR3G* transcript (*Figure 4—figure supplement 1*), we examined the transcription factors that were previously identified to be expressed in root stele (*Brady et al., 2011*). Cross-referencing of these two datasets revealed one transcription factor, *WKRY75* (AT5G13080) that was both enriched in root stele, and potentially binding to *SR3G* promotor.

WRKY75 was previously described to act as a positive regulator of salt stress response by enhancing *SOS1* expression (*Lu et al., 2023*). To investigate whether the *SR3G* expression depends on *WRKY75*, we examined the *SR3G* expression in the *wkry75* T-DNA insertion lines. *SR3G* expression was undetected in *wrky75* knock-down mutants (*Figure 9B*), suggesting that *WKRY75* regulates *SR3G* expression. In comparison, expression of *WRKY75* remained unchanged in the *sr3g-5* mutant when compared to Col-0 (*Figure 9C*). To investigate whether *WRKY75* directly binds to the *SR3G* promoter, we performed Y1H assay. Similar to the two positive controls, the strains transformed with WRKY75-AD and SR3G:pLacZ grew on the SD/-Trp/-Ura+X Gal medium, thereby confirming the interaction between *WRKY75* and *SR3G* promoter region (*Figure 9D*). In contrast, *WRKY75* did not show binding affinity to the promoter region of the adjacent gene to the *SR3G* (i.e. DUF247-150, AT3g50150), thereby reinforcing the uniqueness of the interaction between the *SR3G* and *WRKY75* (*Figure 9E*).

To further examine the functional consequences of the relationship between SR3G and WRKY75, we generated *wrky75-3/sr3g-5* double mutants, and examined their root system architecture alongside the single mutants and Col-0. Under control conditions, both *sr3g-5* and *wrky75-3* mutants showed significant reduction in main root length compared to Col-0 and the two double mutants (*Figure 10A*). Under 75 mM salt, only *wrky75-3* showed significant reduction in main root length compared to all other genotypes tested, confirming the previously described salt sensitivity for *wrky75* mutant under salt stress (*Lu et al., 2023*). We have not observed any differences in main root length under 125 mM salt across all genotypes tested (*Figure 10A*). Regarding lateral root number (*Figure 10B*) and length (*Figure 10C*), we observed significant reduction in lateral root number and length for the two single mutants, *sr3g-5* and *wrky75-3*, compared to other genotypes only under control condition (*Figure 10B–C*). We further investigated Na$^+$ and K$^+$ accumulations as well as Na$^+$/K$^+$ ratio in plate grown seedlings (*Figure 10D*, *Figure 10—figure supplement 1*). At 75 mM salt concentration, a subtle non-significant reduction in root Na$^+$/K$^+$ ratio was observed in the *wrky75* single mutant compared to the other genotypes tested. Conversely, shoot Na$^+$/K$^+$ ratio exhibited a slight non-significant decrease in the two double mutants compared to the other three genotypes at 75 mM salt concentration. At 125 mM salt concentration, root Na$^+$/K$^+$ ratio non-significantly decreased in the two double mutants compared to Col-0 and the two single mutants. Similarly, the two double mutants maintained lower Na$^+$/K$^+$ ratios compared to Col-0 and the two single mutants in shoot at 125 mM salt concentration (*Figure 10D*), indicating that the *SR3G* mutation indeed compensated for the increased Na$^+$ accumulation observed in the *wrky75* mutant under salt stress.

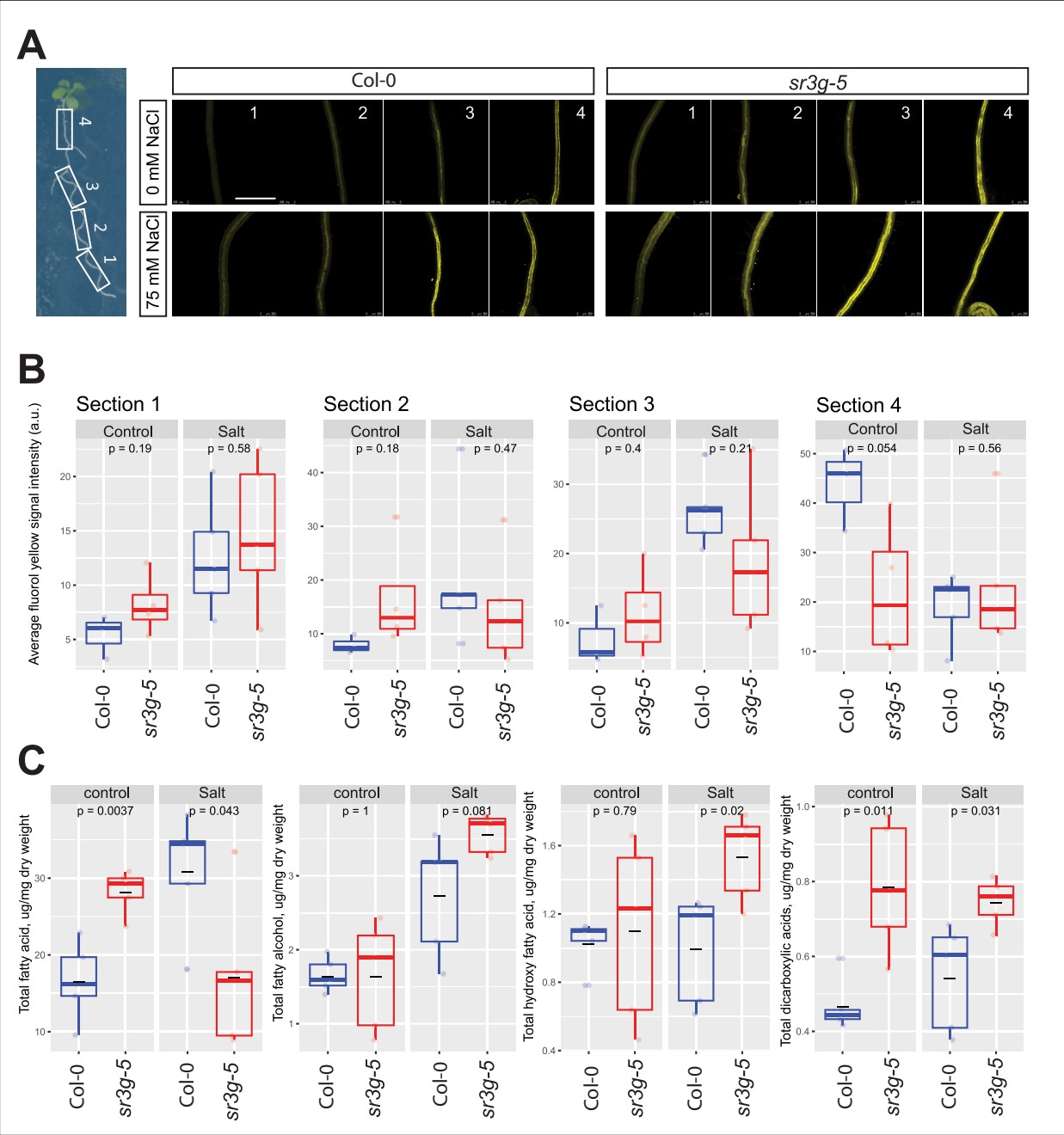

**Figure 8.** *sr3g* mutant root is more suberized than Col-0. (**A**) Representative images of fluorol yellow (FY) staining of four root sections (# 1–4, from root tip to root base as shown by white rectangles across the seedling) were shown for Col-0 and *sr3g-5* mutant with or without 75 mM NaCl. Seeds were germinated on the 1/2 MS plates for 4 d and then seedlings were transferred to the treatment plates with or without 75 mM NaCl for wo more days. (**B**) Quantification of fluorol yellow (FY) signal intensity for each root section. The FY quantification was done using ImageJ. Scale bar is equal to 500 μm in all. Three to five biological replicates were used for FY signal quantification, with n=3 for Col control, n=4 for sr3g control, n=5 for Col under salt stress, and n=5 for sr3g under salt stress. (**C**) Suberin monomers detected using Gas Chromatography–Mass Spectrometry (GC-MS) in the Col-0 and *sr3g-5* mutant roots with or without 75 mM NaCl included fatty acid (FA), fatty alcohol (OH-FA), α,ω -dicarboxylic acids (DCA), and ω -hydroxy fatty acid (ωOH-FA). The significant differences between Col-0 and *sr3g-5* mutant were evaluated using Student 's t-test. Five biological replicates per genotype per condition used for this experiment.

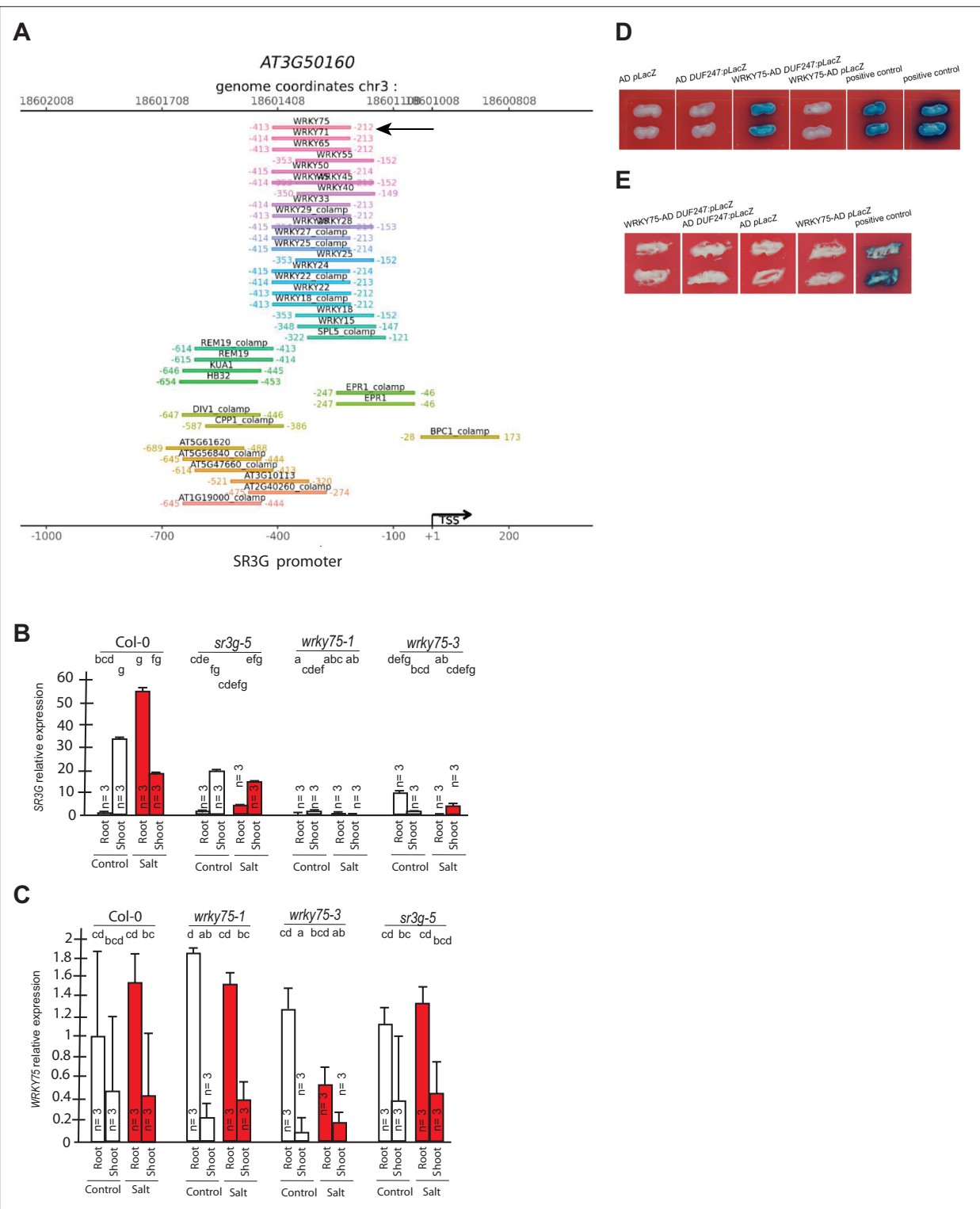

**Figure 9.** WRKY75 binds to the promoter region of the *SR3G* but not to its neighboring DUF247. (**A**) In silico searches in CisCross (https://plamorph.sysbio.ru/ciscross/CCL_index.html) shows potential transcription factors with binding sites on the main *SR3G* (At3g50160) promoter region. A black-arrow points to the *WRKY75* (AT5G13080). (**B**) RT-qPCR showing expression of *SR3G* (AT3G50160) and (**C**) *WRKY75* (AT5G13080) genes, respectively, in Col-0, *sr3g-5* mutant, and two different mutant alleles of the *wrky75*, i.e., *wrky75-1* and *wrky75-3*. RT-qPCR analyses were conducted using seedlings grown on 1/2 x MS for 4 d and then followed by transferring to the treatment plates with or without 75 mM NaCl for 1 wk. Mean values are shown ±SE, with three biological replicates used in each condition and genotype. *AT4G04120* (transposable_element_gene) was used as a reference gene for normalization. Significance was determined by the Tukey–Kramer HSD test in JMP. Levels not connected by same letter are significantly different.

*Figure 9 continued on next page*

*Figure 9 continued*

(**D**) Yeast one-hybrid (Y1H) assay showing *WRKY75* (438 bp) binds to the promoter region of the main *SR3G* (AT3G50160, 953 bp) (**E**) but not to its neighboring promoter, i.e., DUF247-150 (At3g50150, 631 bp). *pB42AD* (AD) and *pLacZ* were used as effector and reporter construct, respectively. Effector and reporter constructs were co-transformed into yeast strain EGY48. Transformants were selected and grown on SD/-Trp-Ura medium. The interactions were observed on SD/-Trp/-Ura+X Gal medium. Empty vector expressing AD domain and *pLacZ* were used as negative control. The two positive controls are NIGT1.4-GolS2 and NIGT1.4-GAE1. The oligo sequences used for Y1H and luciferase assay were provided in ***Supplementary file 4***.

## Discussion

Exposure to abiotic stress alters plant architecture by affecting biomass distribution between individual organs (***Guo et al., 2020***; ***Julkowska and Testerink, 2015***). Exposure to drought alters the distribution of the biomass between root and shoot, to optimize water uptake, while limiting transpiration (***Asch et al., 2005***; ***Chen et al., 2022***; ***Karcher et al., 2008***; ***Uga et al., 2013***). While the processes underlying salt-induced changes in shoot and root growth dynamics and architecture have been systematically described (***Ariel et al., 2010***; ***Awlia et al., 2021***; ***Deolu-Ajayi et al., 2019***; ***Duan et al., 2013***; ***Geng et al., 2013***; ***Julkowska et al., 2017***; ***Zou et al., 2022***), the methodological exploration of biomass distribution between root and shoot for plants exposed to salt stress is underexplored in the current body of research. Here, we developed a tool that allows a non-destructive quantification of root and shoot projected area of the seedlings grown on agar plates (***Figure 1A***). This new tool allows us to efficiently describe changes in biomass accumulation and stress-induced alterations in root and shoot growth. Although the method is restricted to small plants and early developmental stages, it provides valuable insight into stress responses. The application of this tool to an *Arabidopsis* natural diversity panel revealed that salt stress exposure results in loss of coordination between root and shoot growth (***Figure 1D***), rather than a clear directional change, as in the

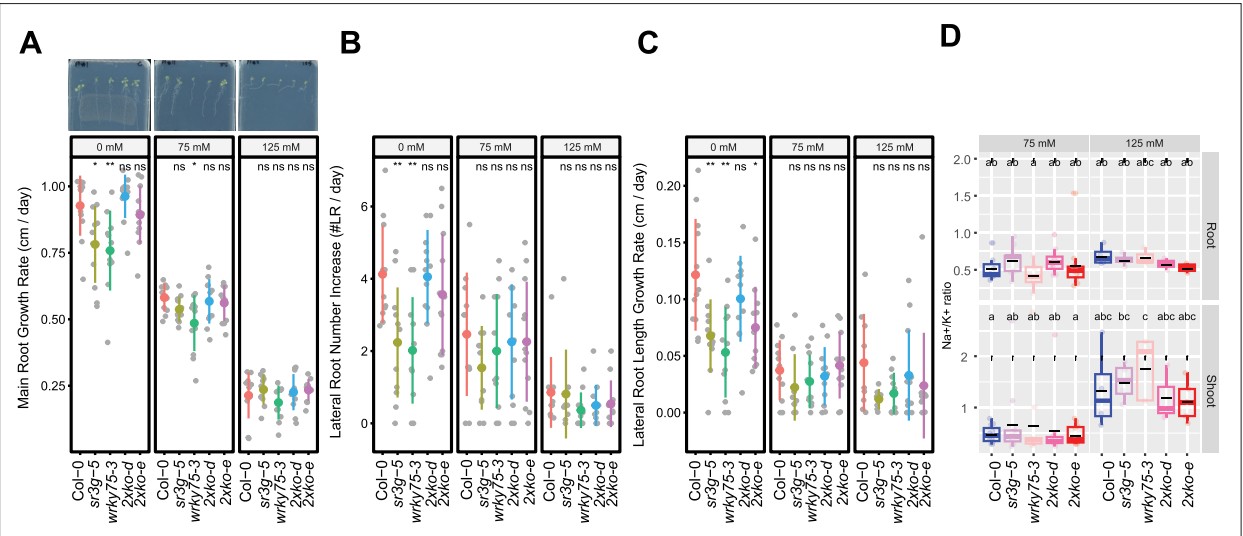

**Figure 10.** *wrky75/sr3g* double mutants roots exhibit reduced sensitivity to salt stress and accumulate lower levels of Na⁺ in their shoots. (**A-C**) Root system architecture analysis for the Col-0, *sr3g-5, wrky75-3,* and two double mutants under various concentrations of NaCl are shown here. (**A**) Main root growth rate, (**B**) Lateral root number increase, and (**C**) Lateral root length increase are shown for the indicated genotypes at various salt concentrations. (**D**) Na⁺/k⁺ ratio in root and shoot of indicated genotypes after 2 wk on treatment plates. Each dot represents individual replicate per genotype. The asterisks above the graphs (**A-C**) indicate significant differences between Col-0 and the genotype, as determined by the Student's t-test: *$p<0.05$ and **$p<0.01$. ns denotes no statistically significant. Statistical analysis in (**D**) was done by comparison of the means for all pairs using Tukey–Kramer HSD test. Levels not connected by the same letter are significantly different ($p<0.05$).

The online version of this article includes the following figure supplement(s) for figure 10:

**Figure supplement 1.** *wrky75/sr3g* double mutants exhibit higher shoot K⁺ contents compared to Col-0 and each individual single mutant in the presence of 125 mM salt.

**Figure supplement 2.** *wrky75/sr3g* double mutants exhibit no noticeable differences in rosette area compared to their individual single mutants when grown in saline soil.

case of drought. To our knowledge, this is the first systematic study of changes in root:shoot ratio. The developed tool for quantifying root and shoot size in plate-grown *Arabidopsis* has the potential to significantly advance research by enabling developmental and stress studies, focusing on coordination of root and shoot development during seedling establishment. Facilitating detailed comparisons of root:shoot ratios, this tool can aid in formulating new hypotheses about the mechanisms contributing to seedling vigor and stress tolerance.

The natural variation in salt-induced changes in root:shoot ratio revealed a clear association with SNPs on chromosome 3 within *SR3G*, that is situated within a region containing eight other *DUF247* genes (*Figure 2*). SR3G was previously associated with self-incompatibility in grasses (*Crain et al., 2020*; *Lian et al., 2021*), as well as with lesions and dwarfism in *Arabidopsis* (*Kondou et al., 2013*) through constitutive activation of defense responses. However, the *SR3G* associated with constitutive defense response was AT3G60470 (*Kondou et al., 2013*), which is located outside the interval associated with salt stress-induced changes in root:shoot ratio (*Figure 2C*). Because we did not observe any consistent phenotypes in the T-DNA insertion lines of *DUF247s* neighboring the *SR3G* (*Figure 6—figure supplements 4–5*), we examined the RNAi lines targeting multiple *DUF247s* including *SR3G* (*Figure 6—figure supplements 6–8*). Our root system architecture analyses revealed no redundancy between *SR3G* and other DUF247s, and the reduced lateral root development in double mutants of RNAi lines simply mimics the phenotype of *sr3g-5* single mutant (*Figure 6*, *Figure 6—figure supplement 6*). Furthermore, we demonstrated that only *SR3G* is regulated by *WRKY75* (*Figure 9B–E*). As these genes are result of tandem duplication, they are expected to accumulate a lot of variation in the cis-regulatory regions. Therefore, differential regulation of two closely related genes can be expected by the accumulated variation within the promoter region.

The results acquired in this study provide multiple layers of evidence that support the role of *SR3G* as a negative regulator of salt stress response in *Arabidopsis*. Loss-of-function *SR3G* maintained higher shoot growth rate and larger rosette area when exposed to salt stress in the soil (*Figure 7A and B*), probably due to reduced $Na^+$ accumulation (*Figure 7D*), which in turn resulted in lower rates of cellular damage (*Figure 7C*). This increased salt tolerance feature can be explained by the increased root suberization (*Figure 7*) as suberin-deficient mutants show susceptibility to salt stress (*Barberon et al., 2016*; *Shukla et al., 2021*; *Ursache et al., 2021*). For example, *Arabidopsis CYP86A1/HORST* (At5g58860) encodes a cytochrome P450-dependent enzyme that is responsible for catalyzing fatty acid $\omega$-hydroxylation. It plays a pivotal role in the biosynthesis of aliphatic suberin, particularly expressed in the root endodermis. Knock-out mutations of *CYP86A1/HORST* result in a substantial decrease of approximately 60% in total root suberin (*Höfer et al., 2008*; *Li et al., 2007*). Consequently, the diminished aliphatic suberin content in the roots of *cyp86a1* mutants leads to increased permeability to water and solutes (*Ranathunge and Schreiber, 2011*). Interestingly, *SR3G* expression is enhanced by salt stress (*Figure 4*, *Figure 4—figure supplement 1*), specifically in the root stele (*Figure 4—figure supplement 1B*). The root stele encompasses vascular bundles, procambium, and pericycle, which gives rise to the secondary growth and lateral root primordia, respectively (*Birnbaum et al., 2003*). The stele is central to the systemic transport of water, minerals, and nutrients within the plant (*Li et al., 2017*; *Ramachandran et al., 2020*; *Zarebanadkouki et al., 2019*), and to the exclusion of salt ions from the transpiration stream (*Møller et al., 2009*).

The deposition of hydrophobic residues of lignin and suberin in the Casparian strips poses a significant barrier to apoplastic transport (*Foster and Miklavcic, 2017*). Previously, silencing genes involved in Casparian strips formation resulted in increased lateral root development (*Wang et al., 2020*). As the suberization of the Casparian strips increases upon salt stress, it limits the accumulation of $Na^+$ and loss of $K^+$ ions from the root stele (*Barberon et al., 2016*). However, the effects of increased root suberization under salt stress have thus far not been described to affect lateral root development. Here, we have identified *SR3G* as a negative regulator of root suberization under salt stress (*Figure 8*), which provides an additional layer of protection against sodium accumulation (*Figure 8*), but negatively impacts lateral root development (*Figure 6A and B*).

As SR3G was previously not described in the context of salt stress, we attempted to identify its molecular context through its transcriptional regulation. Through our in-silico analysis, we have identified *WRKY75* as a putative regulator of *SR3G* expression, and confirmed it using RT-qPCR and Y1H assay (*Figure 9*). WRKY75 was previously described to act as a positive regulator of salt stress tolerance through activation of *SOS1* expression (*Lu et al., 2023*). Similar to the previous study, we also

observed *wrky75* mutants to be more sensitive to salt stress within our experiments (*Figure 10A–C*). Nevertheless, when the *wrky75* mutation was coupled with *sr3g-5*, the roots displayed a phenotype identical to Col-0 (*Figure 10A*), implying that the salt sensitivity of the *wrky75* mutant was counteracted by the *SR3G* mutation. Considering *WRKY75*'s role in positively regulating *SOS1* expression during salt stress, it is plausible that its binding to the *SR3G* promoter inhibits its expression, thereby enhancing salt tolerance on a larger scale.

One of the main players in Na$^+$ transport is High-affinity K$^+$ Transporter (HKT1) (*Laurie et al., 2002*). HKT1 is expressed in root stele (*Mäser et al., 2002*), where it retrieves Na$^+$ from the transpiration stream and limits ion accumulation in the shoot, resulting in higher salt stress tolerance at later developmental stages (*Møller et al., 2009*). At early developmental stages, *HKT1* overexpression reduces the development of lateral root primordia (*Julkowska et al., 2017*), thus creating less Na$^+$ entry gate into the main root, resulting in less Na$^+$ accumulation in shoot. Similarly, *sr3g-5* mutants developed fewer lateral roots that are consistent with the lower Na$^+$ levels in the root upon salt stress (*Figure 6*). The effects of HKT1 and SR3G differ in the magnitude of repression of lateral root development, with HKT1 mutants resulting in almost complete cessation of lateral root development (*Julkowska et al., 2017*), whereas repression of SR3G function has a much more subtle phenotype. Whereas the effect of cell-type specific overexpression of HKT1 on plant performance was negative during the early developmental stage, due to impairing lateral root development, the *sr3g* mutants showed increased salt tolerance when exposed to salt stress treatment during their early vegetative growth (*Figure 7*). These results suggest that reduced lateral root development under salt stress can have a gradient of responses leading to increased and decreased salt tolerance, and the effect is highly dependent on the wider physiological mechanisms, and their contributions to ion exclusion and maintained plant development throughout salt stress exposure.

In conclusion, the study of salt-induced changes in the root:shoot ratio, alongside the development of new tools to quantify these changes during the seedling establishment stage, represents an important new perspective into mechanisms underlying salt stress tolerance. Identification of SR3G as a novel regulator of root suberization, shoot growth, and Na$^+$ accumulation provides better understanding of molecular mechanisms underlying plant responses to salt stress, but also offers a new avenue to explore more nuanced and tissue-specific strategies for enhancing salt stress tolerance.

## Materials and methods
### Screening HapMap population for the increase in root and shoot size
The images collected within (*Julkowska et al., 2017*) were re-analyzed in this manuscript to examine salt-induced changes in the root:shoot ratio for the data presented here. 360 accessions of the *Arabidopsis* HapMap population (*Weigel and Mott, 2009*) were screened over seven experimental batches, using four biological replicates per genotype per condition. The seeds were surface sterilized, incubated in sterile MQ water at 4 °C for 48 hr, and germinated on media containing ½ Murashige-Skoog, 0.5% sucrose, 0.1% MES buffer, pH 5.8 (KOH), and 1% Dashin agar. The 4-d-old seedlings were subsequently transferred to media additionally supplemented with 0, 75, or 125 mM NaCl. The plates were scanned every second day using a flat-bed scanner for eight consecutive days.

The collected images were analyzed for green and white pixels, corresponding to the projected surface of the shoot and root, respectively (https://github.com/ronellsicat/PlantSeg, copy archived at *Sicat, 2024*). A MATLAB-based tool was developed to simplify and speed up the segmentation and analysis pipeline. For automatic segmentation, the tool uses a combination of image operations (histogram equalization), thresholding on different color spaces (e.g. RGB, YCbCr, Lab, HSV), and binary image processing (boundary and islands removal). As the tool is digitalizing various color scales and classifies pixels into either white (root), green (shoot), or blue (background) categories, the adjustment for white balance is obsolete. For analysis, the tool uses connected components analysis to extract the plants (combined roots and shoots) and can guide the user to find possible mistakes which can be manually corrected via the user interface. After segmentation, the number of pixels for roots and shoots are counted and recorded automatically. More details are provided in the publicly available repository for the tool (https://github.com/ronellsicat/PlantSeg, copy archived at *Sicat, 2024*). The collected images and corresponding data acquired from them using the tool can be accessed at Zenodo Repository (experimental batch 1: https://doi.org/10.5281/zenodo.7272155; experimental

batch 2: https://doi.org/10.5281/zenodo.7272344; experimental batch 3: https://doi.org/10.5281/zenodo.7272647; experimental batch 4: https://doi.org/10.5281/zenodo.7271841; experimental batch 5: https://doi.org/10.5281/zenodo.7272139; experimental batch 6: https://doi.org/10.5281/zenodo.7273936; experimental batch 7: https://doi.org/10.5281/zenodo.7273937; quantified data: https://doi.org/10.5281/zenodo.7268673). The tool's precision was evaluated by growing the Col-0 seedlings on agar plates, analyzing their root and shoot projected area on the final day using the tool, and correlating it with the recorded fresh and dry weight of the seedlings (*Figure 1—figure supplement 1*). The collected data from the HapMap panel was analyzed using R for potential outliers (samples identified to be more than 3 x standard deviation from the mean of each genotype within each condition), and the genotype-specific mean was calculated using the stats package (*Team et al., 2018*). The overall increase in root and shoot area was investigated for differences between control and salt stress conditions (*Figure 1—figure supplement 2*). The root and shoot growth were estimated by fitting a linear function for a log-transformed projected root or shoot area over the non-transformed timeline of the experiment (https://rpubs.com/mjulkowska/HapMap-root-shoot-data). The data were subsequently used to explore differences between the experimental batches and the standard deviations within individual accessions. The genotypes above 3 standard deviations from the population's mean were excluded from further analysis. The salt tolerance index (STI) was calculated for root and shoot growth by dividing the average trait value at salt stress by the genotype-specific average at control conditions (*Figure 1—figure supplement 3*). The correlations between individual traits were examined using the genotypic mean (*Figure 1—figure supplement 4*). All quantitative data for root and shoot growth was visualized using ggplot2, corrplot, and ggpubr packages (*Kassambara, 2023*; *Wei and Simko, 2017*; *Wilke, 2016*). For all measured traits, we estimated broad sense heritability (H$^2$) using the MVApp (*Julkowska et al., 2019*; *Supplementary file 1*). The data used as an input for GWAS can be found in *Supplementary file 2*.

## Genome-wide association study

The genotype-specific mean was used as an input to GWAS. The GWAS was conducted as described in *Awlia et al., 2021*, initially using 250 k SNPs (*Horton et al., 2012*) and subsequently using 4 M SNPs (https://1001genomes.org/). The GWAS associations were evaluated for minor allele count (MAC) and association strength above the Bonferroni threshold with -log10(p-value/#SNPs), calculated for each sub-population of SNPs above threshold MAC (*Supplementary file 3*, Bonf.threshold.MAC. specific). The GWAS data was processed for associations with all SNPs used to generate Manhattan and QQ-plots (R notebook for processing GWAS data: https://rpubs.com/mjulkowska/GWAS-root-shoot-Arabidopsis-salt-250kSNPs). Additionally, we prioritized the genetic loci where multiple SNPs were associated with individual traits, or multiple traits were associated with one genetic locus. Our final selection yielded 14 associations with traits measured under control conditions, while 9 and 23 associations were specific to traits measured under 75 and 125 mM NaCl, respectively (*Supplementary file 3*). The GWAS output files can be accessed at Zenodo Repository (GWAS using 250 k SNPs: https://doi.org/10.5281/zenodo.7271751; GWAS using 4 M SNPs: https://doi.org/10.5281/zenodo.7271741). For the most promising loci (*Supplementary file 4*), the sequence information of 147 accessions belonging to the HapMap population was downloaded from the 1001 genomes project website (https://1001genomes.org/) and aligned with ClustalO as described in *Julkowska et al., 2016*. More details on plotting the genome divergence in the loci of interest can be found at https://doi.org/10.17504/protocols.io.t2ieqce. The input and output files of Plot Divergence analysis can be found in Zenodo repository (https://doi.org/10.5281/zenodo.7278021).

## Haplotype analysis

For the most promising candidate loci (*Supplementary file 4*), we have identified the gene open reading frames that were located within the genome-wide linkage-disequilibrium (LD) of the associated SNPs. The LD was expanded if multiple SNPs were identified within the region, and the region of interest was expanded based on the number of coding genes within the LD window. The protein-coding genes were subsequently used for haplotype analysis. All the SNPs located within the gene coding region were used to perform a haplotype analysis. The detailed protocol used for haplotype analysis can be accessed at https://doi.org/10.17504/protocols.io.i2gcgbw. The accessions were grouped into haplotype groups depending on the combination of SNPs within the protein-coding

region of the gene of interest. The haplotypes represented by less than three accessions were excluded from further analysis. The phenotypes of individual haplotypes were explored for significant differences using ANOVA. The results for all investigated haplotypes can be accessed at Zenodo Repository (https://doi.org/10.5281/zenodo.7277703).

## Identification of SR3G orthologs and paralogs

DUF247 domain-containing orthologs from *Arabidopsis thaliana* and eight other species were identified by reciprocal best BLAST within CoGe (*Nelson et al., 2018*; https://genomevolution.org/coge/CoGeBlast.pl) using *SR3G* (At3g50160) nucleotide sequence as query with an E-value cutoff of 1.00E-10. The gene ID, species, and gene names used in this study are listed in *Supplementary file 5*. Sequence homologs were further confirmed by multiple alignments of those hits within each species using Geneious Prime. Sequences with missing or incomplete DUF247 domains were removed, with the exception of the mis-annotated DUF427 homologs in *Camelina sativa* which contained an extraneous intron that, after removal, resulted in a set of intact DUF247 open reading frame containing genes.

## Phylogenetic and positive selection analysis of SR3G orthologs

Identified orthologs were translated into amino acid, then aligned by MUSCLE. Alignments were carefully validated by manual correction. Briefly, diverged insertions were trimmed, shifted frames were adjusted, and conserved regions after translation were retained for phylogeny reconstruction. Phylogenetic reconstruction was performed using the RAxML (*Stamatakis, 2014*) plug-in within Geneious (Geneious Prime 2022.1.1) with the GAMMA-GTR nucleotide model, rapid bootstrapping with a parsimony search, and 100 bootstraps. The distribution of *SR3G* and other DUF247 orthologs, root lengths for each branch, were visualized by the software Geneious Prime.

To characterize any potential positive selection that might exist along certain branches of the inferred phylogeny, including SR3G and its orthologs, we performed a branch-site model A test on branches leading to SR3G, and branches leading to the clade containing SR3G and its closet orthologs using PAML4.0 (*Stamatakis, 2014*; *Yang, 2007*) as performed previously (*Beilstein et al., 2015*; *Nelson et al., 2014*). Specific branches are denoted as $\omega 0$, $\omega 1$, $\omega 2$, $\omega 3$, and $\omega$-rest (Table S???). In the null hypothesis ($H_0$): we assigned a substitution rate ($\omega$=dN/dS) along with $0<\omega<1$ or $\omega$=1 to all branches (represented by $H_0$: $\omega 0 = \omega 1 = \omega 2 = \omega 3 = \omega$-rest). The alternative hypotheses (HAs) here allow for differential selection to be applied to respective branch(es). Specifically, the five alternative hypotheses are listed here: $H_1$: $\omega$-rest branches = $\omega 0 = \omega 3 = \omega 1 \neq \omega 2$, $H_2$: $\omega$-rest branches = $\omega 0 = \omega 1 = \omega 2 \neq \omega 3$, $H_3$: $\omega$-rest branches = $\omega 0 \neq \omega 1 \neq \omega 2 \neq \omega 3$, $H_4$: $\omega$-rest branches $\neq \omega 0 \neq \omega 1 \neq \omega 2 \neq \omega 3$, $H_{sep}$: each branch has its unique $\omega$ value. To test these hypotheses, we further validate or reject alternative hypotheses by conducting a Chi-square test based on likelihood scores (Ln-L score) derived from the respective branch-site model A tests (Chi-square test cutoff: p<0.05).

## Screening candidate genes identified through GWAS

The genes located within the selected loci (*Supplementary file 4*) were further inspected for their contributions to root development and root:shoot ratio under control and salt stress conditions. The T-DNA insertion lines (*Supplementary file 4*) were genotyped using primers listed in *Supplementary file 4*, to confirm the homozygous insertion of the T-DNA within the phenotype lines. All the mutant lines were used for seed propagation in the controlled growth chamber with 16/8 hr of light/dark period, at 22 °C, 60% humidity, and 200 µmole m$^{-1}$ s$^{-1}$ under long-day conditions. The seeds of plants identified as homozygous were subsequently used for the experiment on agar plates, as described before for the Genome-Wide Association Study. A detailed protocol for plate assays can be accessed at https://doi.org/10.17504/protocols.io.zkpf4vn. The plate images were analyzed for root system architecture using Smart Root (*Lobet et al., 2011*) and the root:shoot ratio using the tool described before for Genome-Wide Association Study. The analysis of the mutant lines was performed in R. The data was first curated for selecting possible outliers within a genotype and subsequently used to fit growth functions for root and shoot tissues using exponential functions.

## Screening SR3G mutants within root:shoot locus

The following T-DNA lines (listed in *Supplementary file 4*) were used for SR3G (AT3G50160) as the main GWAS candidate: *sr3g-4* (SAIL_690_E12, glufosinate resistant) and *sr3g-5* (SAIL_608_C06, glufosinate resistant). All T-DNA insertion alleles were in the background of Col-0 accession. The position of T-DNA insertions was verified using PCR (primers listed in *Supplementary file 4*). RNAi lines targeting the neighboring of the *SR3G* or a combination of *SR3G* and its neighboring *DUF247* genes included CSHL#3 targeting DUF130, CSHL#4 targeting SR3G, CSHL#5 targeting DUF130 and DUF170, CSHL#6 targeting DUF130 and DUF140, CSHL#7 targeting DUF10 and DUF150, CSHL#8 targeting DUF140 and DUF190, CSHL#10 targeting DUF180, CSHL#11 targeting DUF150 and DUF170, CSHL#12 targeting DUF130, DUF140, DUF170, DUF180, and DUF190, CSHL#13 targeting DUF140 and DUF170, CSHL#15 targeting DUF140, DUF170, DUF180, and DUF190, CSHL#16 targeting DUF130, SR3G, and DUF170, CSHL#18 targeting DUF150 and DUF190, CSHL#19 targeting DUF130, DUF140, DUF150, SR3G, DUF170, DUF180, and DUF190, CSHL#20 targeting DUF130 and SR3G, CSHL#21 targeting DUF130 and DUF150, and CSHL#22 targeting DUF150 and SR3G. The RNAi lines were verified using RT-qPCR, with the primers listed in *Supplementary file 4*. The following T-DNA insertion alleles (also listed in *Supplementary file 4*) were used to characterize the neighboring genes of the *SR3G* (AT3G50160): 120.2 (SAIL_382_A09), 140.1 (SALK_044685), 140.2 (SALK_005466), 140.3 (SALK_122700), 150.1 (SALK_003824), 150.3 (SALK_071080), 170.1 (SALK_009186), 170.2 (SALK_112602), 170.3 (SALK_008710), 170.4 (SALK_145999), 170.5 (SALK_072937), 190.2 (SALK_137791), 200.1 (SALK_113759), 200.2 (SALK_093701), 200.3 (SALK_129634). The primers used for genotyping PCR are shown in *Supplementary file 4*.

## Screening WRKY75 mutant lines and generation of *wrky75/sr3g* double mutant

The following T-DNA insertion alleles were used for *WRKY75* (AT5G13080): *wrky75-1* (SALK_004954, kanamycin-resistant) and *wrky75-3* (SALKseq_046273, kanamycin-resistant). All T-DNA insertion alleles were in the background of Col-0 accession. The primers used for genotyping PCR are shown in *Supplementary file 4*. Two sets of double mutant combinations were generated for *wrky75-3/sr3g-5*, with *wrky75-3* serving as the maternal parent in one (referred to as line d) and *sr3g-5* serving as the maternal parent in the other (referred to as line e).

## Plasmid construction and plant transformation

For plant expression, all clones were constructed using the GreenGate cloning system (*Lampropoulos et al., 2013*). We have generated full-length SR3G constructs, as well as three fragments - fragment A (from amino acid number 1–216), fragment B (from amino acid number 217–468), and fragment C (from amino acid number 469–503). Fragments A, B, AB, and ABC (full length) were cloned into entry vector pGGC (GreenGate) with mVenus as N-terminal tag. SR3G fragments were PCR amplified using high-fidelity DNA polymerase, and cloned as full-length cDNA or truncated version. The entry clone was evaluated using Sanger sequencing to verify the absence of mutations. Promoters used included Ubiquitin 10 promoter (*Norris et al., 1993*). The list of constructs generated within this project can be found in *Supplementary file 6*. The SR3G constructs generated in this paper and their corresponding vector maps can be accessed in Zenodo repository (https://doi.org/10.5281/zenodo.10788995). The final constructs were validated via vector digestion, using BaeGI for the constructs with fragments A, B, and AB and PvuI for the full length SR3G construct. Subsequently, we transformed them into *Agrobacterium tumerfaciens* strain *GV3101*. Plant transformations were conducted using the floral dip method (*Clough and Bent, 1998*). Transgenic plants were selected for BASTA resistance (10 mg/L) for three generations, to ensure the homozygous and non-segregating population of seedlings for each construct.

## Agar-based plate experiments for mutant lines

*Arabidopsis* seeds were sterilized for 10 min with 50% bleach and rinsed five times using milli-Q water, and germinated on ½ strength Murashige and Skoog (MS) medium containing 0.5% (w/v) sucrose, 0.1% (w/v) 4-morpholineethanesulfonic acid (MES), and 1% (w/v) agar, with PH adjusted to 5.8 with KOH. After 24 hr of vernalization at 4°C in the dark, the plates were placed in the Conviron growth chamber with the light intensity of 130–150 µmol × m$^{-2}$ × s$^{-1}$ in a 16 hr light/8 hr dark cycle at

21°C and 60% humidity. For plate assays, 4 d after germination, the seedlings were transferred to ½ MS media supplemented with 0, 75, or 125 mM NaCl as indicated. The plates were scanned using EPSON scanner every other day, starting from 4 d after germination until the plants were 14 d old. To analyze root system architectural traits from the scanned plate images, we used SmartRoot plugin (*Lobet et al., 2011*) in ImageJ to trace the root, and extracted root-related features in the CSV format followed by data analysis in R (https://rpubs.com/mri23/913282).

## Evaluation of plant performance under salt stress in soil

The seeds were germinated in ½ MS media for 1 wk, as described for the agar-based plate experiments. One week after germination, the seedlings were transplanted to the pot (12×4 cm insert) containing the Cornell Mix soil (per batch combine: 0.16 m³ of peat moss, 20.84 kg of vermiculite, 0.59 kg of Uni-Mix fertilizer, and 2.27 kg of lime) watered to 100% water holding capacity and placed in the walk-in growth chamber with the 16 hr light/8 hr dark period, 22 °C and 60% relative humidity throughout the growth period. When all of the pots dried down to the weight corresponding to 50% of their water holding capacity, they were soaked for 1 hr in tap water or a 200 mM NaCl solution, resulting in an effective concentration of 100 mM NaCl based on the 50% soil water holding capacity, which corresponded to a moderate level of salt stress (*Awlia et al., 2016*). The control pots were soaked for the same length of time in 0 mM NaCl solution, to account for the soil saturation effect. We then allowed the pots to be drained for 2–3 hr to eliminate excess moisture. The pots were placed under phenotyping rigs equipped with an automated imaging system (*Yu et al., 2023*) and the pot weight was measured daily to maintain the reference weight corresponding to 50% of the soil water holding capacity throughout the experiment. We would like to note that this gravimetric-based method for application of salt stress has been developed for soils typically used for pot-grown plants, with relatively high-water holding capacity (*Awlia et al., 2016*). Within these specific conditions, no drought stress symptoms were observed.

To evaluate the growth of soil-grown plants, we used an in-house developed automated imaging system using Raspberry Pi cameras to capture the growth dynamics over the period of 2 wk (*Yu et al., 2023*). The collected images were processed using the PlantCV pipeline (*Gehan et al., 2017*), followed by the data analysis in R, as described here (https://rpubs.com/mjulkowska/sr3g_Maryam2021OctNov).

## Transient expression of SR3G constructs in tobacco epidermal leaf

*Agrobacterium tumefaciens* strain *GV3101* containing cellular marker free GFP (*35 S::GFP* with rifampicin and spectinomycin resistant), generously provided by Professor Maria Harrison Lab at Boyce Thompson Institute, or mVenus-SR3G constructs (including *UBQ10::mVenus-SR3G-full length*, *UBQ10::mVenus-SR3G-fragment A*, *UBQ10::mVenus-SR3G-fragment B*, and *UBQ10::mVenus-SR3G-fragment AB*, all with spectinomycin and tetracycline resistant) were grown in liquid YEP medium with appropriate antibiotics overnight at 28˚C while shaking. The following concentrations were used for antibiotics when used for transient transformation: spectinomycin (50 ug/ml), tetracycline (10 ug/ml), and rifampicin (12.5 ug/ml). Rifampicin and kanamycin (50 ug/ml) were used for P19 plasmid (generously provided by Maria Harrison Lab at BTI) as a silencing inhibitor. The following day, $OD_{600}$ was measured to be 0.3–0.5. The cultures were then centrifuged at 8000 x g for 2 min. The Agrobacterium pellet was resuspended in an induction buffer containing 10 mM $MgCl_2$, 10 mM MES, pH 5.6, and supplemented with 200 µM acetosyringone and was incubated for 2–4 hr in the dark at room temperature. Leaves of 4-wk-old *Nicotiana benthamiana* plants were infiltrated using a syringe. After 48 hr, the fluorescence was visualized in epidermal cells of leaf discs using confocal microscopy.

Confocal microscopy images were taken using the Leica TCS SP5 Laser Scanning Confocal Microscope with the following setting: Excitation at wavelengths of 488 nm (GFP), 514 nm (mVenus), and 561 nm (chlorophyll autofluorescence) were provided using an argon or DPSS (Diode-Pumped Solid State) laser. A spectral emission range of 505–525 nm for GFP, 526–560 nm for mVenus, and 680–700 nm for chlorophyll autofluorescence was used.

## Elemental analysis

The elemental analysis was performed using Inductively Coupled Plasma Atomic Emission Spectroscopy (ICP-AES). For analysis of $Na^+$ and $K^+$ ions in roots and shoots of plate-grown seedlings, the

plants were grown on ½ × MS plates as described above. After 2 wk of exposure to 0, 75, or 125 mM NaCl, root and shoot tissues were harvested and measured for fresh weight, rinsed in milli-Q water, and collected into separate paper bags that were dried at 60°C for 2–3 d. Subsequently, the dry weight was recorded. Samples were digested in double distilled HNO3, followed by the addition of 60/40 nitric/perchloric acid continuing incubation at 150 °C. Samples were processed for ICP-AES analysis using a Thermo iCap 7000 ICP-AES after being diluted to 10 ml with deionized water. For soil-grown plants, the entire rosette leaves were harvested at the end of week four and underwent a similar procedure as described above. Ion content was calculated per dry weight for each sample, followed by data analysis in R (sr3g:https://rpubs.com/mri23/913285, wrky75/sr3g double mutants: https://rpubs.com/mri23/1157092 for plate-grown and sr3g:https://rpubs.com/mri23/913299 for soil-grown seedlings).

## Electrolyte leakage analysis

The electrolyte leakage measurement was done as previously described (*Hu et al., 2015*) with slight modifications. In brief, for each plant, three leaf discs were incubated in 2 ml of distilled water for 24 hr at the light at room temperature with gentle shaking, followed by measuring the initial electrolyte leakage using a conductivity meter (Horiba LAQUAtwin EC-11 Compact Conductivity Meter, # 3999960125). The samples were then subjected to 80°C for 2 hr to release the total electrolyte, and cooled at room temperature for 24 hr. Final electrolyte leakage was measured the following day. The electrolyte leakage percentage was calculated by dividing initial conductivity by final conductivity * 100 and followed by data analysis in R (https://rpubs.com/mri23/913288 and https://rpubs.com/mri23/1102053).

## RT-qPCR analysis

Total RNA was isolated from *Arabidopsis* seedlings root and shoot treated with or without 75 or 125 mM NaCl for 1 wk using the RNeasy Plant Mini Kit (Qiagen). RNA samples were cleaned at least twice using RNase-free DNase (Qiagen) to remove genomic DNA contamination. One microgram of total RNA was used for a reverse transcriptase reaction using iScript cDNA Synthesis Kit (Catalog #170–8891; Bio-Rad). RT-qPCR analysis was conducted using iQ SYBRGreen Supermix (Bio-Rad) according to the manufacturer's instructions in the CFX96 real-time PCR system (Bio-Rad). *AT4G04120* (*transposable_element_gene*) was used as a housekeeping gene for data normalization. RT-qPCR experiments were conducted using three independent experiments, each with at least three biological replicates. The list of primers used is shown in *Supplementary file 4*.

## Suberin staining

Suberin content was examined in 6-d-old *Arabidopsis* roots treated with or without 75 mM NaCl for 2 d using Fluorol Yellow 088 (FY; Santa Cruz Biotechnology Incorporation) staining. Seedlings grown on agar plates (as described above for mutant screens) were incubated in a freshly prepared solution of FY (0.01% [w/v] in lactic acid 85% [W/W]) at 70°C for 30 min, rinsed in milli-Q water for three times with 5 min each, and mounted on slides with 50% glycerol before confocal microscopy examination using excitation at wavelength of 488 nm and a spectral emission range of 500–550 nm. Suberin quantification was performed using the ImageJ software followed by data analysis in R (https://rpubs.com/mri23/935019).

## Suberin monomer quantification using gas chromatography–mass spectrometry (GC-MS)

100–150 *Arabidopsis* seedlings were grown on the plates for four complete days, transferred to 75 mM salt plate for 2 d, and roots were harvested in distilled water to remove any remaining agar. Each plate was treated as one biological replicate, and five plates were used for each genotype/condition. Each plate of roots was collected into a microcentrifuge tube and dried in a speed vacuum concentrator (Thermo Electron, Waltham, MA), and weighed. Approximately 1 mg of dried root were collected from each plate. Suberin monomer quantification was performed according to *Delude et al., 2017*; *Kreszies et al., 2019* with modifications. Dried root tissue was incubated in 1% cellulase plus 1% pectinase in 10 mM citrate buffer (pH 3.0) for 2 wk, with the incubation solution changed every 3 d. After the cellulase/pectinase solution was removed, chloroform/methanol solution (2:1 and 1:2, v/v)

were mixed with the root tissue for 3 d respectively, to remove lipids. The remaining root tissue was mixed with 1 ml freshly prepared 5% $H_2SO_4$ in methanol (v/v) and 200 µl of toluene containing 20 µg of heptadecanoic acid as internal standard, sealed in an 8 ml capped tube and heated at 85 °C for 90 min. After samples were cooled to room temperature, 200 µl hexane and 1.5 ml 0.9% sodium chloride in water are added to the capped tube with shaking. After centrifuging at 1000 $g$ for 1 min, the upper organic phase was transferred to a glass insert of GC vial and dried. 40 µl pyridine was added to the glass insert, 40 µl N,O-bis(trimethylsilyl)trifluoroacetamide (Restek, Bellefonte, PA) was added subsequently, incubated at 65 °C for 2 hr. Standards of 1 µl derivatized sample was injected into GCMS (Agilent 6890–5975, Santa Clara, CA), with the inlet temperature of 250 °C, column helium flow of 1 ml/min, and a split ratio of 5:1. The oven temperature was held at 50 °C for 1 min, raised to 200 °C at the rate of 25 °C/min, held at 200 °C for 1 min, raised to 320 °C at the rate of 10 °C/min, and finally held at 320 °C for 5 min. 1-Docosanol (22:0-OH), 1-eicosanol (20:0-OH), hexadecanedioic acid (16:0-DCA), 1-hexadecanol (16:0-OH), 16-hydroxy hexadecanoic acid (16:0-$\omega$OH), C4 - C24 Even Carbon Saturated FAMEs (Sigma 49,453 U), 20-hydroxy arachidic acid (20:0-$\omega$OH) (Cayman Chemical Company, Ann Arbor, Michigan) standards were input at 2.5 µg, 5 µg, 10 µg, and 20 µg into a mixture and processed with the same procedure as the tested samples to build the standard curve. The MS spectrum of derivatized products with ±2 n carbon number of these standards was calculated and identified, referring to the retention order in *Delude et al., 2017*.

## Yeast one-hybrid assay

The respective combinations of *pB42AD*-fusion effectors and *pLacZ* reporters were co-transformed to yeast strain EGY48 (Shanghai Maokang, China). Transformants were selected and grown on SD/-Trp-Ura medium. The putative interactions were observed on SD/-Trp-/Ura+X Gal after a 3d-incubation at 30 °C. Yeast transformation and growth assay were performed as described in the Yeast Protocols Handbook (BD Clontech, USA). In brief, 631 bp of the *DUF247-150* (AT3G50150) and 953 bp of the *SR3G* (At3g50160) promoter region were separately ligated into *pLacZ* vector as the reporter, while the CDS of *WRKY75* (AT5G13080) was cloned into *pB42AD* vector as the effector. The *SR3G::pLacZ* and *WRKY75-pB42AD* were transformed into the yeast strain EGY48(Shanghai Maokang, China). The respective combinations of *pB42AD*-fusion effectors and *pLacZ* reporters were co-transformed into yeast strain EGY48. Transformants were selected and grown on SD/-Trp-Ura medium. The putative interactions were observed on SD/-Trp/-Ura+X Gal after a 3d-incubation at 30 °C. Yeast transformation and liquid assay were conducted as described in the Yeast Protocols Handbook (BD Clontech, USA). The primers used for the Y1H assay are shown in *Supplementary file 4*.

## Acknowledgements

The authors would like to thank BTI and KAUST Greenhouse Teams for their care of the plants. The authors would like to acknowledge support from the NSF-IOS #2023310 (ADLN) and NSF-IOS #2102120 (ADLN). The majority of funding for this work was generously provided from KAUST baseline funding awarded to Mark Tester, and BTI's startup funds awarded to Magdalena Julkowska.

## Additional information

### Funding

| Funder | Grant reference number | Author |
|---|---|---|
| National Science Foundation | 2023310 | Li'ang Yu Andrew DL Nelson |
| National Science Foundation | 2102120 | Li'ang Yu Andrew DL Nelson |

The funders had no role in study design, data collection and interpretation, or the decision to submit the work for publication.

## Author contributions
Maryam Rahmati Ishka, Data curation, Formal analysis, Validation, Investigation, Writing – original draft, Writing – review and editing, Conceptualization; Hayley Sussman, Data curation, Validation, Investigation, Writing – review and editing; Yunfei Hu, Mashael Daghash Alqahtani, Data curation, Formal analysis, Investigation, Methodology, Writing – review and editing; Eric Craft, Formal analysis, Methodology, Writing – review and editing; Ronell Sicat, Rachid Ait-Haddou, Software, Investigation, Methodology, Writing – review and editing; Minmin Wang, Formal analysis, Investigation, Methodology, Writing – review and editing; Li'ang Yu, Data curation, Formal analysis, Writing – review and editing; Bo Li, Data curation, Formal analysis, Supervision, Funding acquisition, Visualization, Writing – review and editing; Georgia Drakakaki, Resources, Supervision, Funding acquisition, Writing – review and editing; Andrew DL Nelson, Miguel Pineros, Resources, Supervision, Writing – review and editing; Arthur Korte, Resources, Software, Writing – review and editing; Łukasz Jaremko, Formal analysis, Writing – review and editing; Christa Testerink, Resources, Supervision; Mark Tester, Conceptualization, Resources, Supervision, Writing – review and editing; Magdalena M Julkowska, Conceptualization, Data curation, Formal analysis, Supervision, Funding acquisition, Validation, Visualization, Writing – original draft, Writing – review and editing

## Author ORCIDs
Maryam Rahmati Ishka http://orcid.org/0000-0002-5447-643X
Hayley Sussman https://orcid.org/0000-0002-1331-3319
Yunfei Hu https://orcid.org/0009-0004-9799-7798
Eric Craft https://orcid.org/0000-0003-0966-8296
Ronell Sicat https://orcid.org/0000-0001-7037-1614
Minmin Wang https://orcid.org/0000-0001-9545-7895
Li'ang Yu https://orcid.org/0000-0002-9556-011X
Rachid Ait-Haddou http://orcid.org/0000-0002-2606-127X
Bo Li https://orcid.org/0000-0002-3603-3352
Georgia Drakakaki https://orcid.org/0000-0002-3949-8657
Andrew DL Nelson https://orcid.org/0000-0001-9896-1739
Miguel Pineros https://orcid.org/0000-0002-7166-1848
Arthur Korte https://orcid.org/0000-0003-0831-1463
Łukasz Jaremko https://orcid.org/0000-0001-7684-9359
Christa Testerink http://orcid.org/0000-0001-6738-115X
Mark Tester https://orcid.org/0000-0002-5085-8801
Magdalena M Julkowska https://orcid.org/0000-0002-4259-8296

Reviewer #1 (Public review): https://doi.org/10.7554/eLife.98896.4.sa1
Reviewer #2 (Public review): https://doi.org/10.7554/eLife.98896.4.sa2
Author response https://doi.org/10.7554/eLife.98896.4.sa3

## Additional files

### Supplementary files
Supplementary file 1. The individual traits recorded and calculated from the HapMap accessions exposed to the three different levels of salt stress. The root and shoot projected area were quantified using a custom developed tool. The change in the root and shoot area over time was used to calculate Growth Factor (GF) by fitting an exponential function. The broad sense heritability (H2) was calculated for each trait over individual days and conditions using MVApp. This file supplements *Figure 1*.

Supplementary file 2. Genotypic mean data used as an input for genome-wide association study (GWAS). The genotypic mean data was calculated from at least four biological replicates per genotype per condition. The root and shoot area were extracted using a custom developed tool. The Growth Factors (GF) for root and shoot were calculated using exponential growth function. Total seedling size (Tot) was calculated by adding root and shoot area for an individual day. Shoot per Total (SpT) ratio was calculated for individual days by dividing genotype-specific shoot area by the total seedling area for that day and condition. Root per Shoot ratio (RpS) was calculated by dividing genotype-specific root area by the shoot area for that day and condition. This file supplements

*Figure 1*.

Supplementary file 3. Significant associations identified with genome-wide association study (GWAS) of all used traits. The location of individual SNPs is listed according to its location on Chromosome (Chr) and position (Pos). The individual SNP Minor Allele Count (MAC) and Minor Allele Frequency (MAF) is calculated based on specific SNP set used (4 Million or 250 Thousand – listed in 'Mapping' column). The traits associated with individual SNPs are abbreviated as RpS for Root-per-Shoot, SpT for Shoot-per-Total seedling area, Tot for Total seedling area, GR for growth rate, and SHIIT for Shoot Ion Independent Tolerance index (Salt/Control). This file supplements *Figure 2*.

Supplementary file 4. Most important genome-wide association study (GWAS) associations. The most important associations identified through GWAS (*Supplementary file 3*) were inspected further for their functions using either T-DNA or RNAi mutant lines. List of oligos used for genotyping, RT-qPCR, cloning, and Y1H assay are provided. This file supplements *Figure 2*.

Supplementary file 5. The gene ID, species, and gene names used for identification of SR3G orthologs and paralogs. This file supplements *Figure 3*.

Supplementary file 6. Background and foreground values of omega. This file supplements *Figure 3*.

Supplementary file 7. Links to cloning construct maps generated in this study. This file supports *Figure 5*.

MDAR checklist

## Data availability

All data generated or analyzed in this study are available in publicly accessible repositories. The raw images and extracted data used for phenotypic analysis of *Arabidopsis* HapMap accessions have been deposited in Zenodo under the following accession links: Experimental batch 1: https://doi.org/10.5281/zenodo.7272155; Experimental batch 2: https://doi.org/10.5281/zenodo.7272344; Experimental batch 3: https://doi.org/10.5281/zenodo.7272647; Experimental batch 4: https://zenodo.org/records/7271841; Experimental batch 5: https://doi.org/10.5281/zenodo.7272139; Experimental batch 6: https://doi.org/10.5281/zenodo.7273936; Experimental batch 7: https://doi.org/10.5281/zenodo.7273937; quantified data: https://doi.org/10.5281/zenodo.7268673. Genome-Wide Association Study (GWAS) results have been deposited in Zenodo: GWAS using 250k SNPs: https://doi.org/10.5281/zenodo.7271751; GWAS using 4M SNPs: https://doi.org/10.5281/zenodo.7271741. Input and output files from the Plot Divergence analysis are available at https://doi.org/10.5281/zenodo.7278021. The results of haplotype analysis can be accessed at https://doi.org/10.5281/zenodo.7277703. Construct maps for SR3G cloning vectors have been deposited at https://zenodo.org/records/10788996. Scripts and tools used for phenotypic image analysis are available on GitHub: https://github.com/ronellsicat/PlantSeg (copy archived at *Sicat, 2024*). R analysis notebooks for GWAS processing and phenotypic analysis are available at https://rpubs.com/mjulkowska/HapMap-root-shoot-data https://rpubs.com/mjulkowska/GWAS-root-shoot-Arabidopsis-salt-250kSNPs. Additional phenotypic analysis: https://rpubs.com/mjulkowska/sr3g_Maryam2021OctNov. Protocols used in this study are available as follows: haplotype analysis: https://doi.org/10.17504/protocols.io.i2gcgbw; plate assays for stress screening: https://doi.org/10.17504/protocols.io.zkpf4vn; plot divergence analysis: https://doi.org/10.17504/protocols.io.t2ieqce. All other relevant data supporting the findings of this study are provided within the manuscript, supplementary files, and source data files. Further inquiries regarding data access should be directed to the corresponding author.

The following datasets were generated:

| Author(s) | Year | Dataset title | Dataset URL | Database and Identifier |
|-----------|------|---------------|-------------|-------------------------|
| Julkowska MM | 2022 | Arabidopsis HapMap screen for salt-induced changes in root architecture and root:shoot ratio - images BA1 experiment | https://doi.org/10.5281/zenodo.7272155 | Zenodo, 10.5281/zenodo.7272155 |

*Continued on next page*

*Continued*

| Author(s) | Year | Dataset title | Dataset URL | Database and Identifier |
|---|---|---|---|---|
| Julkowska MM | 2022 | Arabidopsis HapMap screen for salt-induced changes in root architecture and root:shoot ratio - images BA2 experiment | https://doi.org/10.5281/zenodo.7272344 | Zenodo, 10.5281/zenodo.7272344 |
| Julkowska MM | 2022 | Arabidopsis HapMap screen for salt-induced changes in root architecture and root:shoot ratio - images BA3 experiment | https://doi.org/10.5281/zenodo.7272647 | Zenodo, 10.5281/zenodo.7272647 |
| Julkowska MM | 2022 | Arabidopsis HapMap screen for salt-induced changes in root architecture and root:shoot ratio - images BA5 experiment | https://doi.org/10.5281/zenodo.7272139 | Zenodo, 10.5281/zenodo.7272139 |
| Julkowska MM | 2022 | Arabidopsis HapMap screen for salt-induced changes in root architecture and root:shoot ratio - images BA6 experiment | https://doi.org/10.5281/zenodo.7273936 | Zenodo, 10.5281/zenodo.7273936 |
| Julkowska MM | 2022 | Arabidopsis HapMap screen for salt-induced changes in root architecture and root:shoot ratio - images BA7 experiment | https://doi.org/10.5281/zenodo.7273937 | Zenodo, 10.5281/zenodo.7273937 |
| Julkowska MM | 2022 | Arabidopsis HapMap screen for salt-induced changes in root:shoot ratio - data for all experimental batches | https://zenodo.org/records/7268673 | Zenodo, 10.5281/zenodo.7268673 |
| Julkowska MM | 2022 | Root:shoot ratio GWAS Arabidopsis 250k SNP mapping | https://doi.org/10.5281/zenodo.7271751 | Zenodo, 10.5281/zenodo.7271751 |
| Julkowska MM | 2022 | Root:shoot ratio GWAS Arabidopsis 4M SNP mapping | https://doi.org/10.5281/zenodo.7271741 | Zenodo, 10.5281/zenodo.7271741 |
| Julkowska MM | 2022 | LocusDivergence analysis of GWAS candidates identified for root:shoot ratio changes under salt stress in Arabidopsis | https://doi.org/10.5281/zenodo.7278021 | Zenodo, 10.5281/zenodo.7278021 |
| Julkowska MM | 2022 | Haplotype analysis of GWAS candidates identified for root:shoot ratio changes under salt stress in Arabidopsis | https://doi.org/10.5281/zenodo.7277703 | Zenodo, 10.5281/zenodo.7277703 |
| Sussman H, Julkowska MM | 2024 | Plasmid Maps of DUF247 Constructs | https://doi.org/10.5281/zenodo.10788995 | Zenodo, 10.5281/zenodo.10788995 |
| Julkowska MM | 2022 | Arabidopsis HapMap screen for salt-induced changes in root architecture and root:shoot ratio - images BA4 experiment | https://doi.org/10.5281/zenodo.7271691 | Zenodo, 10.5281/zenodo.7271691 |

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
