## [Editor Report · eLife Assessment]

Through cellular, developmental, and physiological analysis, this **valuable** study identifies a gene that regulates the relative growth of roots and shoots under salt stress. The holistic approach taken provides **convincing** evidence that this member of a larger tandemly duplicated gene family together with an upstream regulator contributes to salt tolerance. The manuscript will be of interest to plant biologists studying mechanisms of abiotic stress tolerance.

---

## [Referee Report · Reviewer #1 (Public review)]

The authors aim to assess the effect of salt stress on root:shoot ratio, identify the underlying genetic mechanisms, and evaluate their contribution to salt tolerance. To this end, the authors systematically quantified natural variations in salt-induced changes in root: shoot ratio. This innovative approach considers the coordination of root and shoot growth rather than exploring biomass and development of each organ separately. Using this approach, the authors identified a gene cluster encoding eight paralog genes with a domain-of-unknown-function 247 (DUF247), with the majority of SNPs clustering into SR3G (At3g50160). In the manuscript, the authors utilized an integrative approach that includes genomic, genetic, evolutionary, histological, and physiological assays to functionally assess the contribution of their genes of interest to salt tolerance and root development.

Comments on latest version:

The authors have largely addressed my concerns and comments. I have no additional comments for this round of review.

---

## [Referee Report · Reviewer #2 (Public review)]

Summary:

Salt stress is a significant and growing concern for agriculture in some parts of the world. While the effects of sodium excess have been studied in Arabidopsis and (many) crop species, most studies have focused on Na uptake, toxicity and overall effects on yield, rather than on developmental responses to excess Na, per se. The work by Ishka and colleagues aims to fill this gap.

Working from an existing dataset that exposed a diverse panel of *A. thaliana* accessions to control, moderate, and severe salt stress, the authors identify candidate loci associated with altering the root:shoot ratio under salt stress. Following a series of molecular assays, they characterize a DUF247 protein which they dub SR3G, which appears to be a negative regulator of root growth under salt stress.

Overall, this is a well-executed study which demonstrates the functional role played by a single gene in plant response to salt stress in Arabidopsis.

Comments on latest version:

All of the issues that I raised in previous reviews have been addressed by the authors. That said, there are several points that I see have come up in subsequent reviews that remain unresolved.

In response to Reviewer 1, comment 2, regarding changes in expression differences, the authors are misinterpreting simple statistical results. They say that they performed Tukey tests for differences of means, finding, for example, that two means have the same group assignments (in this case, both "c,d") but then argue that "we still observed a clear reduction in WRKY75 transcript abundance." This is not how statistical tests work - we cannot perform a formal test for means and then just do an eyeball test. They also misinterpret the result in which one mean is assigned "b,c,d" results and a second "c,d" - these are statistically overlapping means.

Having said this, I do think that the subtle differences in expression between these different alleles is not critical to the central message of the study. It can be difficult to recapitulate results between labs, much less between different synthetic alleles. I think, in this case, we can let readers decide for themselves whether the reported differences - or lack thereof - is important for follow-up work.

---

## [Author Response]

The following is the authors’ response to the previous reviews.

**Public Reviews:**

**Reviewer #1 (Public review):**
Summary:The authors aim to assess the effect of salt stress on root:shoot ratio, identify the underlying genetic mechanisms, and evaluate their contribution to salt tolerance. To this end, the authors systematically quantified natural variations in salt-induced changes in root:shoot ratio. This innovative approach considers the coordination of root and shoot growth rather than exploring biomass and the development of each organ separately. Using this approach, the authors identified a gene cluster encoding eight paralog genes with a domain-of-unknown-function 247 (DUF247), with the majority of SNPs clustering into SR3G (At3g50160). In the manuscript, the authors utilized an integrative approach that includes genomic, genetic, evolutionary, histological, and physiological assays to functionally assess the contribution of their genes of interest to salt tolerance and root development.Comments on revisions:As the authors correctly noted, variations across samples, genotypes, or experiments make achieving statistical significance challenging. Should the authors choose to emphasize trends across experiments to draw biological conclusions, careful revisions of the text, including titles and figure legends, will be necessary to address some of the inconsistencies between figures (see examples below). However, I would caution that this approach may dilute the overall impact of the work on SR3G function and regulation. Therefore, I strongly recommend pursuing additional experimental evidence wherever possible to strengthen the conclusions.(1) Given the phenotypic differences shown in Figures S17A-B, 10A-C, and 6A, the statement that "SR3G does not play a role in plant development under non-stress conditions" (lines 680-681) requires revision to better reflect the observed data.

Thank you to the reviewer for the comment. We appreciate the acknowledgment that variations among experiments are inherent to biological studies. Figures 6A and S17 represent the same experiment, which initially indicated a phenotype for the sr3g mutant under salt stress. To ensure that growth changes were specifically normalized for stress conditions, we calculated the Stress Tolerance Index (Fig. 6B). In Figure 10, we repeated the experiment including all five genotypes, which supported our original observation that the sr3g mutant exhibited a trend toward reduced lateral root number under 75 mM NaCl compared to Col-0, although this difference was not significant (Fig. 10B). Additionally, we confirmed that the wrky75 mutant showed a significant reduction in main root growth under salt stress compared to Col-0, consistent with findings reported in The Plant Cell by Lu et al. 2023. For both main root length and lateral root number, we demonstrated that the double mutants of wrky75/sr3g displayed growth comparable to wild-type Col-0. This result suggests that the sr3g mutation compensates for the salt sensitivity of the wrky75 mutant.

We completely agree with the reviewer that there is a variation in our results regarding the sr3g phenotype under control conditions, as presented in Fig. 6A/Fig. S17 and Fig. 10A-C. In Fig. 6A/Fig. S17, we did not observe any consistent trends in main root or lateral root length for the sr3g mutant compared to Col-0 under control conditions. However, in Fig. 10A-C, we observed a significant reduction in main root length, lateral root number, and lateral root length for the sr3g mutant under control conditions. We believe this may align with SR3G’s role as a negative regulator of salt stress responses. While loss of this gene benefits plants in coping with salt stress, it might negatively impact overall plant growth under non-stress conditions. This interpretation is further supported by our findings on the root suberization pattern in sr3g mutants under control conditions (Fig. 8B), where increased suberization in root sections 1 to 3, compared to Col-0, could inhibit root growth. While SR3G's role in overall plant fitness is intriguing, it is beyond the scope of this study. We cannot rule out the possibility that SR3G contributes positively to plant growth, particularly root growth. That said, we observed no differences in shoot growth between Col-0 and the sr3g mutant under control conditions (Fig. 7). Additionally, we calculated the Stress Tolerance Index for all aspects of root growth shown in Fig. 10 and presented it in Fig. S25.

To address the reviewer request on rephrasing the lines 680-681 from"SR3G does not play a role in plant development under non-stress conditions" (lines 680-681) statement, this statement is found in lines 652-653 and corresponds to Fig. 7, where we evaluated rosette growth in the WT and sr3g mutant under both control and salt stress conditions. We did not observe any significant differences or even trends between the two genotypes under control conditions, confirming the accuracy of the statement. To clarify further, we have added “*SR3G* does not play a role in rosette growth and development under non-stress conditions”.

(2) I agree with the authors that detecting expression differences in lowly expressed genes can be challenging. However, as demonstrated in the reference provided (Lu et al., 2023), a significant reduction in WRKY75 expression is observed in T-DNA insertion mutant alleles of WRKY75. In contrast, Fig. 9B in the current manuscript shows no reduction in WRKY75 expression in the two mutant alleles selected by the authors, which suggests that these alleles cannot be classified as loss-of-function mutants (line 745). Additionally, the authors note that the wrky75 mutant exhibits reduced main root length under salt stress, consistent with the phenotype reported by Lu et al. (2023). However, other phenotypic discrepancies exist between the two studies. For example, (1) Lu et al. (2023) report that w¬rky75 root length is comparable to WT under control conditions, whereas the current manuscript shows that wrky75 root growth is significantly lower than WT; (2) under salt stress, Lu et al. (2023) show that wrky75 accumulates higher levels of Na+, whereas the current study finds Na+ levels in wrky75 indistinguishable from WT. To confirm the loss of WRKY75 function in these T-DNA insertion alleles the authors should provide additional evidence (e.g., Western blot analysis).

We sincerely appreciate the reviewer acknowledging the challenge of detecting expression differences in lowly expressed genes, such as transcription factors. Transcription factors are typically expressed at lower levels compared to structural or enzymatic proteins, as they function as regulators where small quantities can have substantial effects on downstream gene expression.

That said, we respectfully disagree with the reviewer’s interpretation that there is no reduction in WRKY75 expression in the two mutant lines tested in Fig. 9C. Among the two independent alleles examined, *wrky75-3* showed a clear reduction in expression compared to WT Col-0 under both control and salt stress conditions. Using the Tukey test to compare all groups, we observed distinct changes in the assigned significance letters for each case:

Col/root/control (**cd**) vs wrky75-3/root/control (**cd**): Although the same significance letter was assigned, we still observed a clear reduction in WRKY75 transcript abundance. More importantly, the variation in expression is notably lower compared to Col-0.

Col/shoot/control (bcd) vs wrky75-3/shoot/control (a): This is significant reduction compared to Col

Col/root/salt (cd) vs wrky75-3/root/salt (bcd): Once again, the reduction in WRKY75 transcript levels corresponds to changes in the assigned significance letters.

Col/shoot/salt (bc) vs wrky75-3/shoot/salt (ab): Once again, the reduction in WRKY75 transcript levels corresponds to changes in the assigned significance letters.

To address the reviewer’s comment regarding the significant reduction in WRKY75 expression observed in T-DNA insertion mutant alleles of WRKY75 in the reference by Lu et al., 2023, we would like to draw the reviewer’s attention to the following points:

a) Different alleles: The authors in *The Plant Cell* used different alleles than those used in our study, with one of their alleles targeting regions upstream of the WRKY75 gene. While we identified one of their described alleles (WRKY75-1, SALK_101367) on the T-DNA express website, which targets upstream of WRKY75, the other allele (wrky75-25) appears to have been generated through a different mechanism (possibly an RNAi line) that is not defined in the *Plant Cell* paper and does not appear on the T-DNA express website. The authors mentioned they have received these seeds as gifts from other labs in the acknowledgement ”We thank Prof. Hongwei Guo (Southern University of Science and Technology, China) and Prof. Diqiu Yu (Yunnan University, China) for kindly providing the *WRKY75pro:GUS*, *35Spro:WRKY75-GFP*, wrky75-1, and wrky75-25 seeds. We thank Man-cang Zhang (Electrophysiology platform, Henan University) for performing the NMT experiment”.

However, in our study, we selected two different T-DNAs that target the coding regions. While this may explain slight differences in the observed responses, both studies independently link WRKY75 to salt stress, regardless of the alleles used. For your reference, we have included a screenshot of the different alleles used.

b) Different developmental stages: They measured WRKY75 expression in 5-day-old seedlings. In our experiment, we used seedlings grown on 1/2x MS for 4 days, followed by transfer to treatment plates with or without 75 mM NaCl for one week. As a result, we analyzed older plants (12 days old) for gene expression analysis. Despite the difference in developmental stage, we were still able to observe a reduction in gene expression.

c) Different tissues: The authors of *The Plant Cell* used whole seedlings for gene expression analysis, whereas we separated the roots and shoots and measured gene expression in each tissue type individually. This approach is logical, as WRKY75 is a root cell-specific transcription factor with higher expression in the roots compared to the shoots, as demonstrated in our analysis (Fig. 9C).

Based on the reasoning above, we did work with loss-of-function mutants of WRKY75, particularly wrky75-3. To more accurately reflect the nature of the mutation, we have changed the term "loss-of-function" to "knock-down" in line 717.

The reviewer mentioned phenotypic discrepancies between the two studies. We agree that there are some differences, particularly in the magnitude of responses or expression levels. However, despite variations in the alleles used, developmental stages, and tissue types, both studies reached the same conclusion: WRKY75 is involved in the salt stress response and acts as a positive regulator. We have discussed the differences between our study and *The Plant Cell* in the section above, summarizing them into three main points: different alleles, different developmental stages, and different tissue types.

To address the reviewer’s comment regarding "Lu et al. (2023) report that wrky75 root length is comparable to WT under control conditions, whereas the current manuscript shows that wrky75 root growth is significantly lower than WT": We evaluated root growth differently than *The Plant Cell* study. In *The Plant Cell* (Fig. 5, H-J), root elongation was measured in 10-day-old plants with a single time point measurement. They transferred five-day-old wild-type, wrky75-1, wrky75-25, and WRKY75-OE plants to 1/2× MS medium supplemented with 0 mM or 125 mM NaCl for further growth and photographed them 5 days after transfer. In contrast, our study used 4-day-old seedlings, which were transferred to 1/2 MS with or without 0, 75, or 125 mM salt for additional growth (9 days). Rather than measuring root growth only at the end, we scanned the roots every other day, up to five times, to assess root growth rates. Essentially, the precision of our method is higher as we captured growth changes throughout the developmental process, compared to the approach used in *The Plant Cell*. We do not underestimate the significance of the work conducted by other colleagues in the field, but we also recognize that each laboratory has its own approach and specific practices. This variation in experimental setup is intrinsic to biology, and we believe it is important to study biological phenomena in different ways. Especially as the common or contrasting conclusions reached by different studies, performed by different labs and using different experimental setups are shedding more light on reproducibility and gene contribution across different conditions, which is intrinsic to phenotypic plasticity, and GxE interactions.

*The Plant Cell* used a very high salt concentration, starting at 125 mM, while we were more cautious in our approach, as such a high concentration can inhibit and obscure more subtle phenotypic changes.

To address the reviewer’s comment on "Lu et al. (2023) show that wrky75 accumulates higher levels of Na+, whereas the current study finds Na+ levels in wrky75 indistinguishable from WT," we would like to highlight the differences in the methodologies used in both studies. *The Plant Cell* measured Na+ accumulation in the wrky75 mutant using xylem sap (Supplemental Figure S10), which appears to be a convenient and practical approach in their laboratory. In their experiment, wild-type and wrky75 mutant plants were grown in soil for 3 weeks, watered with either a mock solution or 100 mM NaCl solution for 1 day, and then xylem sap was collected for Na+ content analysis. In contrast, our study employed a different method to measure Na+ and K+ ion content, using Inductively Coupled Plasma Atomic Emission Spectroscopy (ICP-AES) for root and shoot Na+ and K+ measurements. Additionally, we collected samples after two weeks on treatment plates and focused on the Na+/K+ ratio, which we consider more relevant than net Na+ or K+ levels, as the ratio of these ions is a critical determinant of plant salt tolerance. With this in mind, we observed a considerable non-significant increase in the Na+/K+ ratio in the shoots of the wrky75-3 mutant (assigned Tukey’s letter c) compared to the Col-0 WT (assigned Tukey’s letters abc) under 125 mM salt, suggesting that this mutant is salt-sensitive. Importantly, the Na+/K+ ratio in the double wrky75/sr3g mutants was reduced to the WT level under the same salt conditions, further indicating that the salt sensitivity of wrky75 is mitigated by the sr3g mutation.

Based on the reasons mentioned above, we believe that conducting additional experiments, such as Western blot analysis, is unnecessary and would not contribute new insights or alter the context of our findings.

**Reviewer #2 (Public review):**
Summary:Salt stress is a significant and growing concern for agriculture in some parts of the world. While the effects of sodium excess have been studied in Arabidopsis and (many) crop species, most studies have focused on Na uptake, toxicity and overall effects on yield, rather than on developmental responses to excess Na, per se. The work by Ishka and colleagues aims to fill this gap.Working from an existing dataset that exposed a diverse panel of *A. thaliana* accessions to control, moderate, and severe salt stress, the authors identify candidate loci associated with altering the root:shoot ratio under salt stress. Following a series of molecular assays, they characterize a DUF247 protein which they dub SR3G, which appears to be a negative regulator of root growth under salt stress.Overall, this is a well-executed study which demonstrates the functional role played by a single gene in plant response to salt stress in Arabidopsis.Review of revised manuscript:The authors have addressed my point-by-point comments to my satisfaction. In the cases where they have changed their manuscript language, clarified figures, or added analyses I have no further comment. In some cases, there is a fruitful back-and-forth discussion of methodology which I think will be of interest to readers.I have nothing to add during this round of review. I think that the paper and associated discussion will make a nice contribution to the field.

We sincerely appreciate the reviewer’s recognition of the significance of our work to the field.

**Recommendations for the authors:**

**Reviewer #1 (Recommendations for the authors):**
Lines 518-519: The statement that other DUF247s exhibit similar expression patterns to SR3G, suggesting their responsiveness to salt stress, is not fully supported by Fig. S14. Please clarify the specific similarities (and differences) in the expression patterns of the DUF247s shown in Fig. S14, as their expression appears to be spatially and temporally diverse. Additionally, the scale is missing in Fig. S14.

We thank the reviewer. We fixed the text and added expression scales to Figure S14.

Line 684, Fig. 6A should be 7A.

Thanks. It is fixed.

Line 686, Fig. 7A should be 7B.

Thanks. It is fixed.

Lines 721-723: The signal quantification in Fig. 8B does not support the claim that "in section one,..., sr3g-5 showed more suberization compared to Col-0." Given the variability and noise often associated with histological dyes such as Fluorol Yellow staining, conclusions should be cautiously grounded in robust signal quantification. Additionally, please specify the number of biological replicates used in both Fig. 8B and C.

We thank the reviewer for their comments. We believe the statement in the text accurately reflects our results presented in Figure 8B, where we stated “non-significant, but substantially higher levels of root suberization in *sr3g-5* compared to Col-0 in sections one to three of the root under control condition (Fig. 8B).” Therefore, we kept the statement and have included the number of biological replicates in the figure legend.

Lines 731-732: Please provide a more detailed explanation of how the significant changes in suberin monomer levels align with the Fluorol Yellow staining results, and clarify how these findings support the proposed negative role of SR3G in root suberization.

Fluorol Yellow is a lipophilic dye widely used to label suberin in plant tissues, specifically in roots in this study. Given the inherent variability in histological assays, we confirmed the increase in suberization using an alternative method, Gas Chromatography–Mass Spectrometry (GC-MS). Both approaches revealed elevated suberin levels in the sr3g mutant compared to Col-0. Since the overall suberin content was higher in the mutant under both control and salt stress conditions, we proposed that SR3G acts as a negative regulator of root suberization.

Lines 686-688 and Figure S24: The authors calculated water mass as FW-DW. A more standard approach for calculating water content is (FW-DW)/FW x 100. Please update the text or adjust the calculation accordingly. Additionally, if the goal is to test differences between WT and the mutant within each condition, a t-test would be a more appropriate statistical method.

We thank the reviewer. We added water content % to the figure S24. We kept the statistical test as it is as we wanted to be able to observe changes across conditions and genotypes.

Lines 633-635 states that "No significant difference was observed between sr3g-4 and Col-0 (Fig. S18), except for the Stress Tolerance Index (STI) calculated using growth rates of lateral root length and number." However, based on the Figure S18 legend and statistical analysis (i.e., ns), it appears that the sr3g-4 mutant shows no alterations in root system architecture compared to Col-0. Please revise the text to accurately reflect the results of the statistical analysis.

We thank the reviewer. We now fixed the text to reflect the result.

Lines 698-707: The statistical analysis does not support the reported differences in the Na+/K+ ratio for the single and double mutants of sr3g-5 and wrky75-3 (Fig. 10D, where levels connected by the same letters indicate they are not significantly different). Furthermore, the conclusion that "the SR3G mutation indeed compensated for the increased Na+ accumulation observed in the wrky75 mutant under salt stress" is also based on non-significant differences (Fig. S25B). Please revise the text to accurately reflect the results of the statistical analysis. Additionally, since each mutant is compared to the WT, I recommend using Dunnett's test for statistical analysis.

We thank the reviewer for their feedback. We have carefully revised the text to better support our findings. As previously mentioned, variations among samples are evident and are well-reflected across all our datasets. We have presented all data and focused on identifying trends within our samples to guide interpretation.

We observed that the SR3G mutation effectively compensated for the increased Na+ accumulation observed in the wrky75 mutant under salt stress. A closer examination of the shoot Na+/K+ ratio under 125 mM salt shows that the wrky75 single mutant has a higher Na+/K+ ratio (indicated by the letter "c") compared to Col-0 (indicated by "abc") and the two double mutants (also indicated by "abc"). Therefore, we have retained the statistical analysis as originally conducted, and maintain our conclusions as is.

Figure 6: data in panel C present the Na/K ratio, not Na+ content. Based on the statistical analysis of root Na+ levels presented in Fig. S17C, there is no significant difference between sr3g-5 and WT. Please update the title of Fig. 6. In addition, in panel A, the title of the Y-axis and figure legend should be "Lateral root growth rate" without the word length, and in panel C, the statistical analysis is missing.

We thank the reviewer. We updated Fig. 6 title and fixed the Y-axis in panel A, and added statistical letters to panel C. Legend was updated to reflect the changes.

Figure 7: Please clearly label the time points where significant differences between genotypes are observed for both early and late salt treatments. Was there a significant difference recorded between WT and sr3g-5 on day 0 under early salt stress? Such differences may arise from initial variations in plant size within this experiment, as indicated by Fig. 7B, where significant differences in rosette area are evident starting from day 0. Additionally, please indicate the statistical analysis in panel E.

We thank the reviewer for this suggestion. We updated the figure with a statistical test added to the panel E. Although the difference between sr3g mutant and Col-0 is indeed significant in its growth rate at day 0, we would like to draw the attention of the reviewer that this growth rate was calculated over the 24 hours after adding salt stress. Therefore, this difference in growth rate is related to exposure to salt stress. Moreover, the growth rate between Col-0 and sr3g mutant does not differ in two other treatments (Control and Late Salt Stress) further supporting the conclusion that sr3g is affecting rosette size and growth rate only under early salt stress conditions.

We have also added the Salt Tolerance Index calculation to Figure S24 as additional evidence, controlling for potential differences in size between Col-0 and sr3g mutant.

Figure S17: statistical analysis is not indicated in panels A, B, and D.

We thank the reviewer for spotting that. We updated the figure with a statistical test.

Figures S21-23: The quality of these figures is insufficient, hindering the ability to effectively interpret the authors' results and main message. Furthermore, a Dunnett's test, rather than a t-test, is the appropriate statistical method for this analysis.

We thank the reviewer for this observation. We have now added a high resolution figures for all supplemental figures, which should increase the resolution of the figures. As we are comparing all of the genotypes to Col-0 one-by-one - the results of individual t-tests are sufficient for this analysis.